# Temporal-Difference Learning Using Distributed Error Signals

**Jonas Guan**[1,2]**, Shon Eduard Verch**[1]**, Claas Voelcker**[1,2]**, Ethan C. Jackson**[1]**,
Nicolas Papernot**[1,2]**, William A. Cunningham**[1,2,3]

[1]University of Toronto      [2]Vector Institute
[3]Schwartz Reisman Institute for Technology and Society

## Abstract

A computational problem in biological reward-based learning is how credit assignment is performed in the nucleus accumbens (NAc) to update synaptic weights. Much research suggests that NAc dopamine encodes temporal-difference (TD) errors for learning value predictions. However, dopamine is synchronously distributed in regionally homogeneous concentrations, which does not support explicit credit assignment (like used by backpropagation). It is unclear whether distributed errors alone are sufficient for synapses to make coordinated updates to learn complex, nonlinear reward-based learning tasks. We design a new deep Q-learning algorithm, ARTIFICIAL DOPAMINE, to computationally demonstrate that synchronously distributed, per-layer TD errors may be sufficient to learn surprisingly complex RL tasks. We empirically evaluate our algorithm on MinAtar, the DeepMind Control Suite, and classic control tasks, and show it often achieves comparable performance to deep RL algorithms that use backpropagation.

## 1 Introduction

Computer science and neuroscience have enjoyed a longstanding and mutually beneficial relationship. This synergy is exemplified by the inception of artificial neural networks, which drew inspiration from biological neural networks. Neuroscience also adopted temporal-difference (TD) learning [62] from reinforcement learning (RL) as a framework for biological reward-based learning in the midbrain [23, 59]. At the intersection of these ideas, deep RL has much benefited from and contributed to interdisciplinary progress between the two fields [45, 11].

An interesting problem raised in biological learning is how signals transmitted by the neuromodulator dopamine computationally induce coordinated reward-based learning. In the mesolimbic system, dopamine is synthesized by dopamine neurons in the ventral tegmental area (VTA) and transmitted through the mesolimbic pathway to several regions, including the nucleus accumbens (NAc). There, it is synchronously distributed in regionally homogeneous concentrations [58], and serves as a reward prediction error signal for synaptic adjustments via TD learning [9, 23].[1] Figure 1 shows a conceptual illustration: the medium spiny neurons in the NAc receive error signals distributed locally in their region via dopamine. Computationally, however, this theory faces the credit assignment problem [25]: the individual synaptic updates using just local errors must somehow work in coordination to improve the collective prediction.[2] Are distributed error signals alone sufficient to coordinate neurons to learn complex reward-based learning tasks?

---

[1]NAc dopamine also serves many roles beyond signaling reward prediction errors [18]; its full responsibilities are an active area of research. We only focus on its role in error signaling, which is most pertinent to our problem.

[2]For clarity, this is different from the *temporal* credit assignment problem, oft discussed in RL literature.

Deep RL typically solves the credit assignment problem using backpropagation (BP) [57]. BP propagates the global error backwards through the network, and computes the gradient (*w.r.t.* the global error) of each layer's synaptic weights sequentially via the chain rule. In contrast to the synchronously distributed errors in the NAc, BP involves neurons sequentially communicating error signals with each other. This sequential propagation explicitly coordinates learning, but also creates dependencies: each layer's updates depend on the error of subsequent layers. This is known as the update locking problem, which is biologically implausible [52], limits parallelization, and cannot explain how distributed error signals may support coordinated learning.

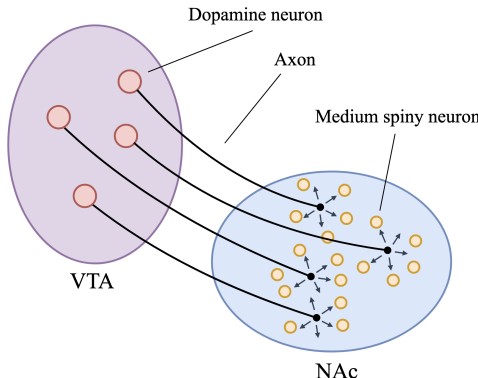

Figure 1: Simplified illustration of dopamine distribution in the NAc. Dopamine is synthesized in the VTA and transported along axons to the NAc, where it is picked up by receptors in medium spiny neurons. Dopamine concentrations (error signals) are locally homogenous, but can vary across regions. Connections between NAc neurons not shown.

Recent ML research on more biologically plausible alternatives to BP may offer critical insights. PEPITA [21] and Forward-Forward (FF) [30] both replace BP's backward learning pass with a second forward pass to address update locking. Most relevantly, Hinton [30] made a surprising discovery: layers can learn useful representations for subsequent layers even when trained independently of the errors of those subsequent layers. In FF, each layer generates its own prediction and error, and is only trained to learn hidden representations that minimize the local error. The subsequent layer takes these representations as input, and achieves better performance over training, despite being unable to send errors to the previous layer. This improves the collective global prediction without explicit, sequential coordination of error signals. To the best of our knowledge, these learning principles have not been explored in RL or biological reward-based learning.

Drawing a novel connection, we hypothesize that the computational mechanisms that enable FF's independent, per-layer training may also enable distributed error signals to support coordinated reward-based learning. To test our hypothesis, we design ARTIFICIAL DOPAMINE (AD), a new deep Q-learning algorithm that trains RL agents using only synchronously distributed, per-layer TD errors, and evaluate its performance on a range of discrete and continuous RL tasks. This provides a potential explanation for credit assignment in NAc dopaminergic learning at the algorithmic level of analysis [42]. Our results show that AD can solve many common RL tasks often as well as deep RL algorithms that use backpropagation, despite not propagating error signals between layers. Thus, we computationally demonstrate that distributed errors alone may be sufficient for coordinated reward-based learning.

AD networks inherit several ideas from FF, which differ from traditional neural networks in two significant ways. First, each layer in an AD network computes its own prediction and receives a corresponding error (Section 3.1). This per-layer error mirrors the locally homogenous distribution of dopamine, and the computation of error and updates can be synchronously parallelized across layers; there are no dependencies across layers. Second, we use forward[3] connections in time to send activations from upper to lower layers (Section 3.2). This provides an information pathway for upper layers to communicate with lower layers using activations, rather than error signals, and empirically improves performance. Figure 2 outlines our architecture, unfolded in time.

The AD cell (Figure 3) is where we differ most significantly from FF. Its role is to compute the local Q-prediction and TD error. FF is designed to separate real from fake (generated) data, a binary classification task. But the NAc is theorized to predict value, a regression task, and therefore needs more precision. To achieve this, we introduce an attention-like mechanism for non-linear regression without using error propagation (Section 3.1).

We evaluate AD on 14 discrete and continuous RL tasks from the MinAtar testbed [70], the DeepMind Control Suite (DMC) [64], and classic control environments implemented in Gymnasium [65].

---

[3]"Forward" and "backward" are widely used in deep learning literature both to describe direction in time and in the order of layers. This can be confusing. For the remainder of this paper, we use "forward/backward" when describing time, and "upper/lower" when describing position among layers, as shown in Figure 2.

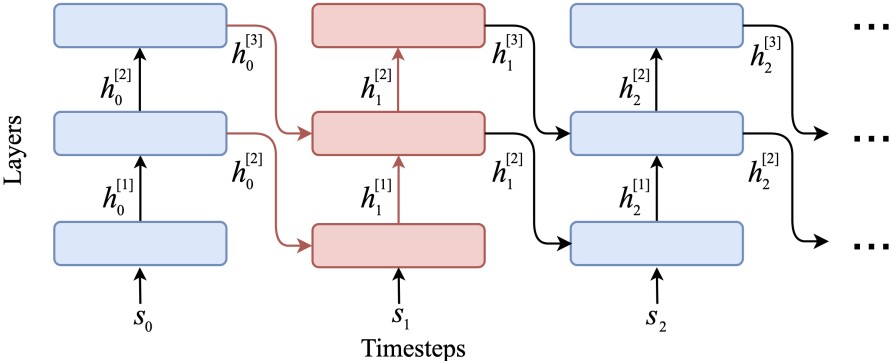

Figure 2: Network architecture of a 3-layer AD network. $h_t^{[l]}$ represents the activations of layer $l$ at time $t$, and $s_t$ the input state. The blocks are AD cells, as shown in Figure 3. Similar to how dopamine neurons compute and distribute error used by a local region, each cell computes its own local TD error used by its updates; errors do not propagate across layers. To relay information, upper layers send activations to lower layers in the next timestep. For example, red shows all active connections at $t = 1$.

MinAtar tasks are miniaturized versions of Atari games, and DMC contains continuous control tasks with simulated physics. These environments are complex enough to reflect many challenges in modern RL [14], yet remain adequately tractable so as not to necessitate extra components like convolutional layers, which may be confounding when attributing performance. We benchmark AD against DQN [45], SAC [26], and TD-MPC2 [28] baselines, and conduct ablation studies to examine the effects of the forward connections and additional layers. Our results in Figures 4 and 8 show that AD learns to solve many of these tasks with comparable performance to the baselines, using just per-layer TD errors. Our code is available at `https://github.com/social-ai-uoft/ad-paper`.

To summarize our core contributions:

- Are distributed TD error signals sufficient to solve credit assignment and learn complex reward-based learning tasks? We construct a computational example of an RL agent that learns using only distributed, per-layer TD errors. This provides evidence that dopamine-distributed signals alone may be enough to support reward-based learning in the nucleus accumbens.

- We design a Q-learning algorithm, ARTIFICIAL DOPAMINE, to train our agent. Like Forward-Forward, AD does not propagate error signals across layers. Unlike FF, we introduce a novel cell architecture to compute Q-value predictions, as Q-learning is a regression task.

- We evaluate our agent on 14 common RL tasks in discrete and continuous control, and show that AD can often achieve comparable performance to deep RL algorithms, without backpropagation.

## 2 Background

### 2.1 Reward-Based Learning in the NAc[4]

The nucleus accumbens (NAc) plays a critical role in the mesolimbic reward system, and is theorized to predict action value [41]. The predicted action value is encoded via the firing rate of NAc neurons [23], which reach dopamine neurons in the ventral tegmental area (VTA) via projections and influence their activity [60]. In addition, the VTA receives reward signals from sensory inputs, such as the detection of sugar on the tongue [23]. The value predictions and reward signals enable the VTA to compute reward prediction errors, *i.e.* differences between expected and actual reward. These reward prediction errors (more specifically TD errors) are encoded via dopamine, and projected through the mesolimbic pathway to regions of the ventral striatum, including the NAc, and lead to synaptic adjustments [60]. When there is a positive reward prediction error, the average activity of dopamine neurons increases; when there is a negative error, the average activity decreases [9].

---

[4]We give a simplified overview of reward-based learning in the NAc and its connections to TD learning here to provide intuition; this is not a complete picture of biological reward-based learning. There exists competing theories for alternative mappings of the TD learning framework to the brain's reward circuit; see Section 6.

The exchange of value predictions and error signals between the NAc and VTA sets the stage for TD learning. TD models have shown strong congruency with observed dopamine activity [48], and are widely established as the primary theory for mesolimbic dopaminergic learning.

What is less known is how NAc neurons computationally use the dopamine-encoded error signals to coordinate learning. Dopamine-encoded error signals are distributed [58], which makes credit assignment difficult, and there are no other known mechanisms that NAc neurons utilize to communicate error signals for explicit coordination. We illustrate the distribution process in Figure 1. Each dopamine neuron projects dopamine along its axon through the mesolimbic pathway, then synaptically releases it in synchronous bursts to the immediate juxtasynaptic area. This causes dopamine concentration in the region to peak, activating dopamine receptors in NAc medium spiny neurons. Dopamine concentration levels are locally homogeneous near the synapses [58], but can vary across different regions of the NAc [68], as dopamine neurons need not all fire at once. This supports the distribution of localized error signals. While Figure 1 is clearly not to scale, dopamine neurons are indeed significantly larger than medium spiny neurons. Their large cell body size enables them to support large terminal fields [13], allowing each dopamine neuron to widely distribute its signal to groups of NAc neurons.

## 2.2 Temporal-Difference Learning

Temporal-difference learning (TD learning) methods are a family of RL approaches that aim to solve the problem of estimating the recursively defined Q function [62]. In their most basic form, TD learning methods can be implemented as lookup tables, where an estimate of the Q function is kept for each state-action pair. However, in many practical applications with large or continuous state spaces, a tabular representation of the Q values for all distinct state-action pairs is computationally infeasible. In these cases, function approximation, for example via linear regression or neural networks, is necessary to obtain an estimate of the Q function. These methods, called Fitted Q Iteration [4] or Deep Q Learning [44], use the squared temporal difference error as a regression loss $\mathcal{L}(s, a, s') = \delta(s, a, s')^2$, where $\delta$ is the TD error, and update a parametric Q function approximation via gradient descent. To prevent the double sampling bias and other instabilities, only $Q(s, a)$ is updated and the next states value is estimated with an independent copy $\bar{Q}$ that is not updated. This is commonly called the bootstrapped Q loss. Given parameters $\theta$, this loss can be written as:

$$\mathcal{L}(s, a, s', \theta) = \left( Q_\theta(s, a) - \left[ r(s, a) + \gamma \max_{a'} \bar{Q}_\theta(s', a') \right] \right)^2$$

For a more formal description of our TD learning setting, see Appendix A.

## 2.3 Forward-Forward

The Forward-Forward (FF) algorithm [30] is a greedy multi-layer learning algorithm that replaces the forward and backward passes of backpropagation with two identical forward passes, positive and negative. The positive pass is run on real data; the negative on fake (generated) data. The goal of the model is to learn to separate real from fake data. During training, each layer of the network performs this classification independently using a measure called *goodness*, which computationally acts as the logit for this binary classification task. Each layer computes its own per-layer error for updating. FF's de facto measure of a layer's goodness $g$ is the sum of its squared hidden activations $h_i$ minus some threshold $\tau$, i.e. $g = \sum_i h_i^2 - \tau$. However, due to the simple goodness formula, if each layer directly passes its activations to the next, the next layer often trivially learns the identity function. It then uses the sum of its last layer's hidden activations as its goodness. This traps the layer in a local optimum. To avoid this, FF uses layer normalization [5] to normalize the activations before passing them on, which keeps the relative values of the activations, but makes them sum to 0.

Compared to BP, a global algorithm that requires the entire network to be updated sequentially, FF is a local algorithm that updates each layer independently via local, layer-wise losses. Critically, the authors of [30] showed that layers can learn useful representations that help subsequent layers even though they are trained on local errors, without explicit credit assignment. However, for a feedforward architecture, one glaring limitation is that later layers cannot relay any information to earlier ones. Hinton suggests addressing this with a multi-layer recurrent architecture, where for each layer $l$ in the network, the input is determined by output of the layer $l - 1$ and that of $l$ and $l + 1$ at the previous time step $t - 1$. This allows for top-to-bottom information flow through the network via activations, which is more biologically plausible.

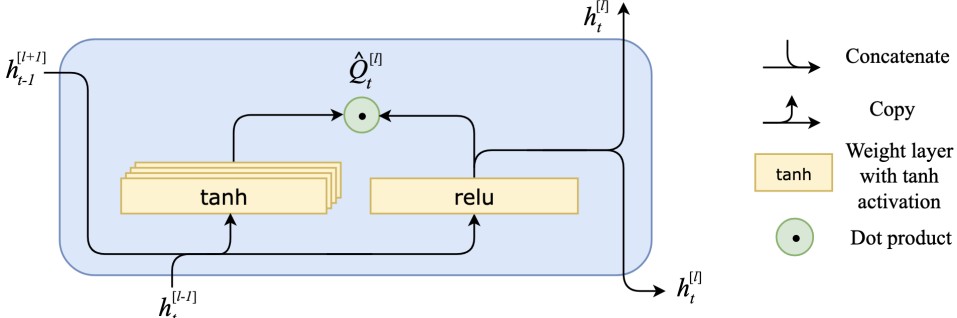

Figure 3: Inner workings of our proposed AD cell (*i.e.*, hidden layer). $h_t^{[l]}$ is the activations of the cell $l$ at time $t$, and $\hat{Q}_t^{[l]}$ is a vector of Q-value predictions given the current state and each action. We compute the cell's activations $h_t^{[l]}$ using a `ReLU` weight layer, then use an attention-like mechanism to compute $\hat{Q}_t^{[l]}$. Specifically, we obtain $\hat{Q}_t^{[l]}$ by having the cell's `tanh` weight layers, one for each action, compute attention weights that are then applied to $h_t^{[l]}$. Each cell computes its own error.

## 3   Artificial Dopamine

ARTIFICIAL DOPAMINE (AD) is a deep Q-learning algorithm that trains deep RL agents using distributed, per-layer TD errors. AD inherits the per-layer predictions and local computation of errors from FF, and adopts the forward-in-time downward connections from FF's recurrent variant. Informally, the intuition is that the per-layer error acts similarly to the locally homogeneous distribution of dopamine in the NAc. The per-layer errors can be computed in a parallelized, distributed fashion; each layer computes its own local error, which the neurons of the layer use to adjust weights according to their contributions to the error. Similarly, the NAc neurons near the synapses of each dopamine neuron receive the same error signal (encoded via dopamine), which they use to adjust their synaptic weights, according to their previous activity shortly before receiving the error. During inference, the network uses the average Q-values across the layers to produce a final prediction.

Since the NAc is theorized to predict action value [23], *i.e.* Q-value, our prediction task deviates from that of FF, which performs binary classification. In general, predicting action value is a nonlinear regression task. To learn this task using a neural network, without resorting to the biologically implausible BP, we design an attention-like mechanism that learns sets of "attention" weights— one set per action head, which introduces nonlinearity. We encapsulate this process in AD cells.

### 3.1   AD Cell Internals

Our network architecture is composed of layers of AD cells, each of which makes its own Q-value prediction, computes its own local TD error, and updates its own weights. At inference, the final prediction of the network is the average of each cell's Q-value predictions. We use an attention-like mechanism [6] that learns a weighted sum of the cell's hidden activations to predict the Q-value. The hidden activations are passed to other cells, but the attention weights and Q-value are withheld. This mechanism simply serves to functionally simulate the complex nonlinear capabilities of biological neurons [10]; we are not attempting to draw an analog between our mechanism and any biological counterpart, and design choices are primarily made based on empirical performance.

The concept of a cell is reminiscent of the design of a recurrent neural network. In our case, there is a single cell per layer, so we use the terms somewhat interchangeably for clarity of exposition. The vital difference is that no loss information is propagated between cells via BP; that is, there is no BP through time. Instead, the same BP are passed to the cell above (i.e., hidden layer) at the same timestep and to the layer below at the next timestep. In the absence of BP, these activations provide a pathway for upper cells to communicate information to lower cells (see Figure 2). We discuss in more detail how these connections operate in Section 3.2.

Figure 3 presents the cell design. Each cell is mechanistically identical and takes in two inputs: the hidden activations of the cell below at the current timestep (or observation, if lowest cell), and the hidden

activations of the cell above at the previous timestep. It produces two identical outputs, sent to the cell above immediately, and the cell below at the next time step. At the start of an episode, the activations from the previous timestep are zeroes. The top cell only receives one input and produces one output.

The attention-like mechanism for Q-value prediction works as follows. Each cell computes its hidden activation, $h_t^{[l]}$, using the layer above's hidden activations from the previous timestep, $h_{t-1}^{[l+1]}$, and the layer below's hidden activations at the current timestep, $h_t^{[l-1]}$. Specifically, it passes the concatenation of $h_{t-1}^{[l+1]}$ and $h_t^{[l-1]}$ through a `ReLU` weight layer (shown in Figure 3) to get $h_t^{[l]}$. The `ReLU` weight layer multiplies the concatenated input $[h_t^{[l-1]}, h_{t-1}^{[l+1]}]$ by its learned weight matrix $W^{[l]}$, then applies the `ReLU` nonlinearity function. In parallel, the cell also uses $[h_t^{[l-1]}, h_{t-1}^{[l+1]}]$ to compute "attention" weights, by multiplying it with the learned weight matrix $W_{\text{att}}^{[l]}$ then applying the tanh function. Finally, the cell takes the dot product between the output of the `tanh` layers, $\tanh(W_{\text{att}}^{[l]} \cdot [h_t^{[l-1]}, h_{t-1}^{[l+1]}])$, and the hidden activations $h_t^{[l]}$, to compute the cell's Q-value prediction $\hat{Q}^{[l]}(s_t, a)$. Each cell reuses its internal weights over time; for example the matrix $W_{\text{in}}^{[1]}$ is used at each timestep in the first cell. Therefore, the full computation performed by a cell is:

$$h_t^{[0]} = s_t, \quad h_t^{[L+1]} = 0, \quad h_{t-1}^{[l]} = 0$$

$$h_t^{[l]} = \text{relu}\left( W_{\text{in}}^{[l]} \cdot \left[ h_t^{[l-1]}, h_{t-1}^{[l+1]} \right] \right)$$

$$\hat{Q}^{[l]}(s_t, a) = \tanh\left( W_{\text{att}}^{[l]} \cdot \left[ h_t^{[l-1]}, h_{t-1}^{[l+1]} \right] \right)^{\mathsf{T}} h_t^{[l]}$$

Optionally, like [30], our architecture supports skip connections between cells. In such cases, the additional input from a skip connection is simply concatenated with $[h_t^{[l-1]}, h_{t-1}^{[l+1]}]$ before computing $h_t^{[l]}$.

### 3.2   Network Connections

As shown in Figure 2, each cell passes its state to the cell $l + 1$ above at the current timestep $t$, and to the cell $l - 1$ below in the next timestep $t + 1$. The information flow is strictly unidirectional to match the direction of time flow in RL environments. This is necessary as interacting with the environment happens sequentially, meaning future information will not be available when acting.

Although we do not backpropagate gradients across cells, information does flow from upper layers to lower layers via the temporal connection (forward in time). The upper layers use the connections to communicate with lower layers via activations, which is more biologically plausible [49]. Our results in Figure 4 suggest that these connections can greatly increase network performance in the absence of BP. The intuition for adopting these forward-in-time connections is that they are well-suited to take advantage of the temporal structure of the Q-values of trajectories for better learning. Given a good policy, the Q-value predictions of a well-trained model should remain stable through each state of a trajectory (assuming the dynamics are reasonably deterministic). This means that the Q-value prediction of the current timestep, and the hidden activations used to make this prediction, can often still be useful for predicting the Q-value of the next timestep. In contrast, in FF's experiments on image classification, this effect is forced– FF repeats the same input every timestep, reducing computational efficiency. Our results empirically support the effectiveness of forward connections for Q-learning, particularly in more complex environments.

## 4   Experiments

The main goal of our experiments is to evaluate whether distributed, per-layer TD errors are sufficient for learning complex RL tasks that are typically only solved with backpropagation and sequential errors. Our criterion of sufficiency is how well the agent learns to solve the given task, measured in terms of average episodic return. Since there are no official standards that define solving these environments, we use the performance of established RL algorithms (*i.e.* DQN, TD-MPC2, and SAC) as the gold standard to compare against. These algorithms are commonly used to solve the environments we choose and are known to be strong performers [70, 28, 26].

Like Hinton [30], our aim is to investigate the learning capabilities of a novel algorithm that operates under additional biological constraints, rather than pursue state-of-the-art performance. Thus, we opt

for a simple implementation with few extensions. The only extensions we employ are experience replay [45] and the Double-Q learning trick [66]– which are standard for deep Q-learning– and skip connections from the input to upper layers for the DMC environments. We do not use convolutional layers, as these more closely resemble the brain's visual cortex [40], and similar structures are not found in the mesolimbic system. We implement our algorithm in Jax [12].

**Training Process.** The training process of our RL agent is based on the standard DQN training algorithm with experience replay [45]. During training, we use a randomly sampled replay buffer that stores every transition observed from the environment. We replay sequences of short length, similar to how other recurrent architectures are trained [29, 35], and compute the local updates for each cell sequentially according to the network graph. Since the local updates are distributed and per-layer, they can be computed in parallel. We provide a formal description of our training process in Appendix B.

**Environments.** We run our experiments on 2 RL benchmarks and 4 classic control tasks, totaling 14 tasks. MinAtar [70] is a simplified implementation of 5 Atari 2600 games: Seaquest, Breakout, Asterix, Freeway, and Space Invaders. The DeepMind Control (DMC) Suite [64] is a set of low-level robotics control environments, with continuous state spaces and tasks of varying difficulty. For our experiments, we used a discretized action space, following Seyde et al. [61], as our architecture is currently only developed for discrete Q-learning. From the DMC tasks we select Walker Walk, Walker Run, Hopper Hop, Cheetah Run, and Reacher Hard. In addition, we provide results on the classic control tasks Cart Pole, Mountain Car, Lunar Lander, and Acrobot, which we include in Appendix C. For a more elaborate discussion on these environments and our task choice, see Appendix I.

**Baselines.** On MinAtar, we compare our results against a fully-connected DQN to make performance comparisons more direct and informative. As the original baselines presented in Young and Tian [70] used a CNN, we replaced the CNN with fully connected layers and tuned hyperparameters for fairness of comparison. We find that the new architecture performs as well as or better than the one presented by Young and Tian [70]. Specifically, we use a double DQN with 3 hidden layers of 1024, 512, and 512 units with `ReLU` activations. This baseline achieves strong performance and has a number of trainable parameters comparable to our networks.

For the continuous control tasks, we show that our method almost reaches the performance of state-of-the-art algorithmic approaches such as SAC [27] and TD-MPC2 [28], which rely on backpropagation. The results were taken from Yarats and Kostrikov [69] and Hansen et al. [28] respectively. We do not change the underlying architectures or hyperparameters.

**Network architecture and hyperparameters.** On MinAtar, we use a 3-hidden-layer network with forward activation connections. The cell output sizes are 400, 200, and 200. Due to additional connections within cells and from upper to lower cells, this architecture has a similar number of trainable parameters as the DQN. On DMC, we use a smaller network with cell output sizes 128, 96, and 96, and discretize the action space following Seyde et al. [61]. For more details, see Appendix G. For each benchmark, we use the same network and hyperparameters across all tasks to test the robustness of our architecture and learning algorithm.

## 5 Results

We present the results of AD on MinAtar and DMC environments in Figure 4, and compare its performance against DQN, SAC, and TD-MPC2. Figure 4 shows the mean episodic return over episodes across 10 random seeds, with standard error. We also provide 95% bootstrap confidence intervals [55] in Appendix D and aggregate statistics in Appendix E. We additionally perform ablation studies by removing the forward connections to lower layers, and measuring the performance of a single-layer AD cell. We show both the forward connections and multiple layers contribute to performance (Figures 5 and 6). Finally, we evaluate an implementation of AD that learns distributions over Q-values, based on recent work by Dabney et al. [20], and find that AD shows promising results (Figure 7).

**Comparison against baselines.** We find that AD is able to learn stable policies reliably in all test environments. On MinAtar tasks, our agent achieves comparable performance to DQN on Breakout, and slightly surpasses DQN's performance on Asterix and Freeway, while DQN performs better on

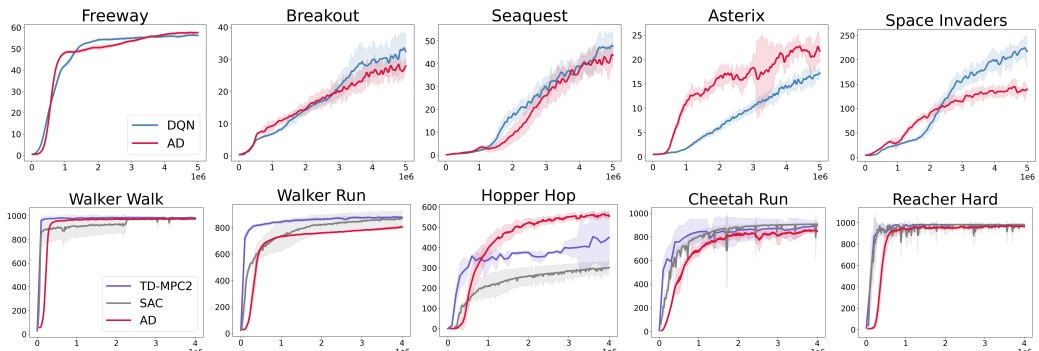

Figure 4: Episodic returns of AD in MinAtar and DMC environments, compared to DQN, TD-MPC2 and SAC. Lines show the mean return over 10 seeds and the shaded area conforms to 3 standard errors. The axes are return and environmental steps.

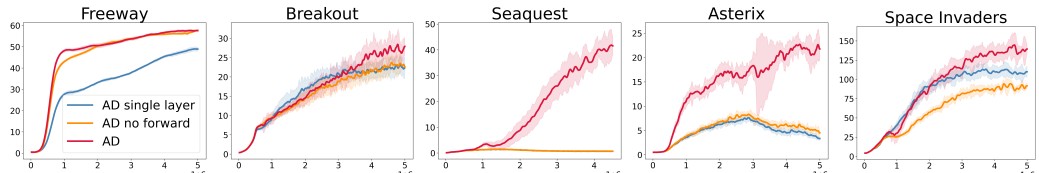

Figure 5: Ablation study comparing the performance of AD against AD without the forward-in-time connections, and a single-layer AD cell. In Seaquest and Asterix, AD achieves qualitatively stronger performance. In Seaquest the line for AD single layer is overlapped by the line for AD no forward.

Seaquest and Space Invaders. On all evaluated DMC tasks, we see that our method's results are close or on par with those of SAC and TD-MPC2 when using both backward connections and multiple layers, however, sample efficiency is marginally lower.

We note that the action space discretization we utilize for DMC tasks may complicate the comparison, as it both slows down our training but can benefit algorithms (compare Seyde et al. [61]). In addition, we do not show results on the hardest DMC tasks because of the difficulties in scaling our approach to large action spaces, which leads to rapid growth in network parameters.

Overall, our results demonstrate that AD shows biological distributed error signals may allow for coordinated learning in several RL domains. Therefore, further refining and improving the architecture for state-of-the-art RL benchmark performance may be an exciting and promising direction for future work.

**Forward connections to lower layers.** To further measure the performance impact of the forward connections in time, we compared the temporal forward version of the network to a 3-layer AD network without the forward-in-time connections. All other hyperparameters are the same as the 3-layer AD network. As shown in Figure 5, this resulted in a moderate drop in performance in most environments, increased variance in the training, and devastating drops in performance on Seaquest and Asterix. These tasks are the most complex out of the five. We provide additional discussion on AD's performance in Seaquest, where the performance difference is most significant, in Appendix N. These results suggests that the information channels from the upper to lower layers are vital for performance on many tasks. On this same note, AD's ability to achieve similar performance to DQN when the forward connections are added suggest that forward connections may be an effective replacement for backpropagation in certain tasks.

**Single-layer performance.** Another important question about our proposed architecture is whether the cells learn to coordinate with each other. A concern is that the majority of the learning may be accomplished by the lowest cell, If the performance of the multi-layer AD network does not improve over the single cell, it would suggest that we cannot explain AD's performance as a result of the cells coordinating their learning using distributed errors. We show in Figure 5 that the multi-layer version of AD outperforms the single-layer at all tasks, and the single-layer cell fails at Seaquest and Asterix.

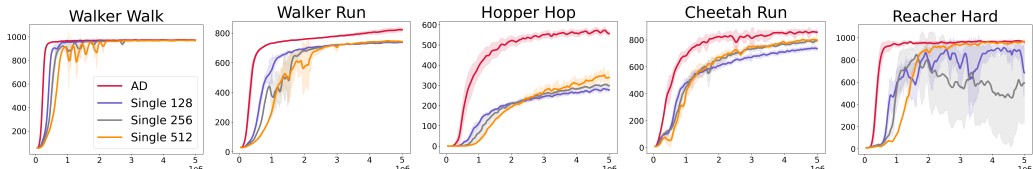

Figure 6: Episodic returns of different-sized single-layer AD, compared to the standard 3-layer AD. Single 128 is a single-layer with 128 hidden activations. Overall, increasing the layer size of the single layer does not result in clear increases in performance. Lines show the mean return over 8 seeds and the shaded area conforms to 3 standard errors. The axes are return and environmental steps.

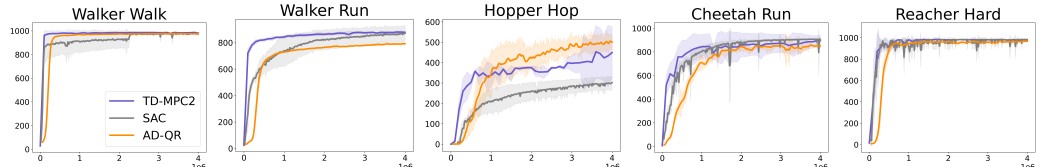

Figure 7: Episodic returns of the distributional RL version of AD, implemented with Quantile Regression (QR). Lines show the mean return over 8 seeds and the shaded area conforms to 3 standard errors. The axes are return and environmental steps.

An additional concern along these same lines is whether a wider, single-layer AD cell may achieve the same level of performance as multi-layer AD. In Figure 6, we show that increasing the layer size of a single-layer AD cell does not result in clear increases in performance in DMC tasks. We also experimented with increasing the layer size of a single-layer cell for Seaquest and Asterix from 400 to 600 and 800, and did not find noticeable improvements in either case.

**Distributional RL.** Recent work by Dabney et al. [20] suggest that the brain's value predictors may be learning distributions over future values, rather than just the mean, as previously believed. Dabney et al. [20] argue that different dopamine neurons in the VTA may have different scalings for positive and negative reward prediction errors– intuitively, they can be more optimistic or pessimistic– which results in the predictors learning a distribution over the values. Interestingly, this coincides with the development of distributional RL, whose algorithms aim to learn such distributions.

To better align our work with the findings of Dabney et al. [20], we additionally implement a version of AD that learns distributions over values, and evaluate it on the DMC tasks. Our implementation is based on Quantile Regression (QR) DQNs [19], and requires just a simple modification to each AD cell. Rather than predicting a single Q-value, each cell predicts 10 Q-values that each match to one quantile of a 10-quantile QR-DQN. The tradeoff is that this requires additional compute.

Our results in Figure 7 suggest that AD may be well-suited for distributional learning. In each of the tasks, our agent achieves similar performance to the standard version of AD, and only slightly lags behind on Hopper Hop. This may be a result of the greater sparsity of the Hopper environment, which makes it more difficult for distributional RL algorithms to learn.

# 6   Limitations

We proposed an algorithm that trains an RL agent using only distributed TD errors, which provides a computational example of how distributed error signals may be sufficient to solve credit assignment in reward-based learning in the NAc. However, our model does not accurately capture all aspects of the relevant biology. In our model, we use per-layer TD errors as an analogy for dopamine neurons distributing error signals to local regions around their synapses. But unlike in artificial neural networks, which form the basis of our architecture, neurons in the NAc are not clearly organized into layers. AD's hierarchical message passing architecture is a design choice we inherit from deep learning practices, and not meant as a mechanistic model of the NAc. Furthermore, activations in biological neurons are communicated asynchronously, and can form recurrent loops, which we do not account for.

In addition, we make some assumptions regarding biological reward-based learning that are not yet conclusive in neuroscience. Most importantly, we assume that neurons in the NAc learn to output action values, and dopamine neurons in the VTA receive reward signals and the predicted action values to compute and return TD errors.[5] While these assumptions are widely supported by research [23, 41, 46, 47], there exist other theories and empirical results that provide alternative explanations for mesolimbic reward-based learning. Some of these works include Roesch et al. [56], who suggest that on-policy value-based learning (SARSA) better explains dopamine activity than off-policy value-based learning (*e.g.* Q-learning); Takahashi [63] and Chen and Goldberg [16], who provide evidence that map subregions of the striatum to actor-critic models; Ito and Doya [32] and Weglage et al. [67] who suggest the dorsal striatum signals action values, whereas the ventral striatum signals state values; and Akam and Walton [2] and Coddington et al. [17], who respectively propose how dopamine may be used by the brain to perform model-based and policy-based learning rather than just value-based learning. Furthermore, while there is strong evidence that some form of TD learning is used by the brain, mappings between RL frameworks and dopaminergic learning may not be mechanistically accurate even if they are behaviorally accurate.

There are also several technical limitations of our work. First, like other Q-learning algorithms, AD requires discrete action spaces. To solve the DMC tasks, which have continuous action spaces, we discretized the action space, following the method used by Seyde et al. [61] (Section 4). While this is consistent with other work in RL, it introduces additional complexities [61] and may not reflect biology. Second, within each AD cell, the number of `tanh` weight layers grows in proportion to the size of the action space (Figure 3). This limits the scalability of AD for tasks with large action spaces. We can mitigate the effects of this issue using a matrix decomposition trick described in Appendix F, but AD currently cannot scale to very large action space DMC tasks like Humanoid or Dog [64]. Third, computational constraints limited the number of runs we perform per task. Additional experiments can further improve the robustness and generalizability of our results.

Finally, our work isolates one system of biological learning, and attempts to provide a computational explanation without accounting for other systems of learning, for example hippocampal contributions to value-based learning [7]. But the brain's learning capabilities are likely a result of combining signals from several systems, which may not be divisible [50], and the NAc may be just one part of a larger value-based system [8]. Unlike backpropagation, the brain utilizes multiple types of learning in conjunction, both supervised and unsupervised, using local and global signals. AD only models one type of learning, *i.e.* error-driven learning using distributed reward signals to induce local updates. Other biologically plausible algorithms, such as ANGC [53], may provide explanations for other forms of learning, which may be critical to building a more complete understanding of biological reward-based learning. Indeed, aspects of Hebbian learning or active inference are likely critical to achieving a fully biologically plausible, efficient, and powerful learning system. In that light, we explore learning with just distributed error signals not to rule out the importance of other methods, but to demonstrate that this one principle alone may be sufficient to solve some complex RL tasks nearly as well as BP. We believe that a key to achieving general, human-like intelligence will be the integration of these different learning methods and learning signals; this is an exciting direction we aim to explore in future work.

## Acknowledgements

We gratefully acknowledge our sponsors, who support our research with financial and in-kind contributions: Amazon, Apple, CIFAR through the Canada CIFAR AI Chair program and the Canadian Foundation for Innovation, DARPA through the GARD project, Meta, NSERC through the Discovery Grant and funding reference number RGPIN-2018-05946, the Ontario Early Researcher Award, the Sloan Foundation, and the Schwartz Reisman Institute for Technology and Society. Resources supporting this research were also provided by the Province of Ontario, the Government of Canada through CIFAR, and the sponsors of the Vector Institute. We extend our sincere gratitude to Mete Kemertas for offering insights that significantly accelerated the initial exploration of this work. We also thank Congyu Fang, David Glukhov, Stephan Rabanser, Anvith Thudi, Sierra Wyllie, and other members of the CleverHans and SocialAI labs, as well as our anonymous reviewers for invaluable discussions and feedback.

---

[5]Note that AD does not assume action selection need happen in the NAc. Rather, we just take the assumption that the NAc computes the Q-value function, which takes in the state and action as parameters, and performs Q-learning. The action may be provided from another region like the dorsal striatum.

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

## A  Formal Definition of TD Learning

We consider a standard discounted infinite horizon Markov Decision Process (MDP) setting with states $\mathcal{S}$, actions $\mathcal{A}$, a transition kernel $p(s'|s, a)$, a reward function $r : \mathcal{S} \times \mathcal{A} \to \mathbb{R}$ and a discount factor $\gamma \in [0, 1]$. This is a typical setting for reinforcement learning [62].

The goal of TD learning is to obtain a policy $\pi(a|s)$ that maximizes the discounted future sum of rewards. The value function $Q : \mathcal{S} \times \mathcal{A} \to \mathbb{R}$ measures how valuable a given action $a \in \mathcal{A}$ is in a given state $s \in \mathcal{S}$, and can be used to directly compute a policy. It is defined via a recursive formula $Q(s, a) = r(s, a) + \gamma \sum_{s'} p(s'|s, a) \sum_{a'} \pi(a'|s')Q(s', a')$. When experience in the form of $s, a, r, s'$ tuples becomes available from interaction with the MDP, the Q function estimates are updated using the TD error $\delta(s, a, s') = r(s, a) + \gamma \sum_{a'} \pi(a'|s')Q(s', a') - Q(s, a)$. The update rule is $Q_{k+1}(s, a) \leftarrow Q_k(s, a) + \alpha_k \delta(s, a, s')$, where $\alpha$ specifies the learning rate.

## B  Training Process

---

**Algorithm 1** AD Q-Learning

---
Initialize replay memory $\mathcal{D}$ to capacity $\mathcal{N}$
Initialize action-value functions $Q^{[1]}, \ldots, Q^{[L]}$ with random weights $\theta^{[1]}, \ldots, \theta^{[L]}$
Initialize target function $\hat{Q}^{[1]}, \ldots, \hat{Q}^{[L]}$ with weights $\theta^{[i]'} = \theta^{[i]}$ for every $i \in \{1, \ldots, L\}$
**for** episode 1, $M$ **do**
$\quad$ Initialize $s_1 =$ initial state
$\quad$ **for** $t = 1, T$ **do**
$\quad\quad$ With probability $\varepsilon$ select a random action $a_t$
$\quad\quad$ otherwise select $a_t \in \arg\max_a \frac{1}{L} \sum_{l=1}^{L} Q^{[l]}(s_t, a; \theta^{[l]})$
$\quad\quad$ Execute action $a_t$ in emulator and observe reward $r_{t+1}$ and next state $s_{t+1}$
$\quad\quad$ Store experience $(s_t, a_t, r_{t+1}, s_{t+1})$ in $\mathcal{D}$
$\quad\quad$ Sample minibatch of $K$-sized episodic sub-trajectories

$$(s_{j-K}, a_{j-K}, r_{j-K+1}, s_{j-K+1}), \ldots, (s_j, a_j, r_{j+1}, s_{j+1})$$

$\quad\quad$ from $\mathcal{D}$ with uniform randomness
$\quad\quad$ Replay states $s_{j-K}, \ldots, s_{j-1}$ through online network to obtain layer activations $h_{j-1}^{[1]}, \ldots, h_{j-1}^{[L]}$
$\quad\quad$ Set current layer input $x \leftarrow s_j$
$\quad\quad$ **for** layer $l = 1, \ldots, L$ **do**
$\quad\quad\quad$ Set TD target $Y_j^{[l]} = \begin{cases} r_{j+1} & \text{if episode terminates at step } j+1 \\ r_{j+1} + \gamma \max_a \hat{Q}^{[l]}\left(s_{j+1}, Q^{[l]}(s_{j+1}, a; \theta^{[l]}); \theta^{[l]'}\right) & \text{otherwise} \end{cases}$
$\quad\quad\quad$ Set prediction $y_j \leftarrow Q^{[l]}(s_t, a_j; \theta^{[l]})$ using $x$ and last activations $h_j^{[l-1]}$ and $h_{j-1}^{[l+1]}$
$\quad\quad\quad$ Perform a gradient descent step on $(Y_j^{[l]} - y_j)^2$ with respect to the layer parameters $\theta^{[l]}$
$\quad\quad\quad$ Set input for next layer $x \leftarrow h_j^{[\ell]}(x)$
$\quad\quad$ **end for**
$\quad\quad$ Every $C$ steps reset $\theta^{[i]'} = \theta^{[i]}$ for every $i \in \{1, \ldots, L\}$
$\quad$ **end for**
**end for**

---

## C  Experiments on Classic Control Environments

Beyond MinAtar and DMC, we also evaluate AD on 4 classic control environments: Cart Pole (also known as inverted pendulum), Mountain Car, Lunar Lander, and Acrobot, and compare its performance against DQN. We show our results in Figure 8.

The main purpose of these experiments is to further support AD's robustness across different RL environments, and indirectly compare its performance against Active Neural Generative Coding (ANGC) [53], another biologically-inspired deep RL algorithm that provides a more plausible

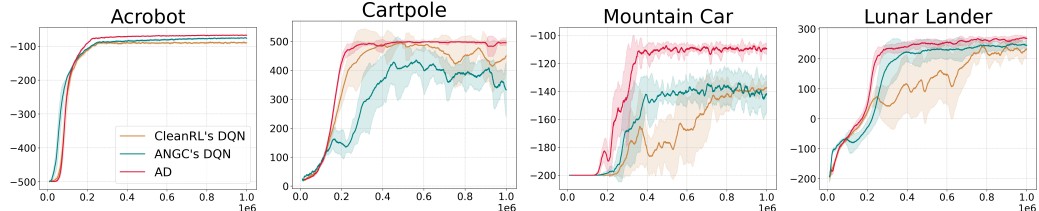

Figure 8: Episodic returns of AD, CleanRL's DQN, and ANGC's DQN. Lines show the mean episode return over 10 seeds and the shaded area conforms to 3 standard errors of the mean.

alternative to backpropagation. Ororbia and Mali [53] evaluates ANGC in 4 environments: Cart Pole, Mountain Car, Lunar Lander, and a custom robot-arm-reaching environment.

In addition, we implement 2 DQNs with different hyperparameters as baselines for comparison: first is CleanRL's reference DQN for classic control problems [31], and second is the DQN tuned by [53] for comparison against their ANGC agent, which we refer to as ANGC's DQN. The CleanRL DQN provides a well-established, publicly-vetted baseline, whereas ANGC's DQN serves as a reference point for us to indirectly compare AD's performance with ANGC.

To better demonstrate AD's robustness across environments, we use the same set of hyperparameters for each of the environments. Specifically, we use a 2-layer AD network, with output sizes 128 and 96, a fixed learning rate of $2.5e^{-4}$ with Adam optimization, an exploration fraction of 0.2, and final epsilon of 0.05. All other hyperparameters are the same as our network for MinAtar, shown in Appendix G. As neither CleanRL nor ANGC's DQN uses a learning rate scheduler, we also removed ours for better comparison. CleanRL's DQN is a 2-layer network with 120 and 84 units; the same DQN is used for all 4 environments. ANGC uses different 2-layer DQNs for each of their environments, which they tuned for environment-specific performance. Their Cart Pole DQN has 256, 256 hidden units; Mountain Car has 256, 192, and Lunar Lander 512, 384. For Acrobot, we used the same hyperparameters as the Cart Pole DQN, given the similarity of the environments, which worked well. For more hyperparameter details, we refer the reader to [31] and [53]; both works present excellent detail.

We made two adjustments when reproducing CleanRL and ANGC's DQNs. First, since AD utilizes double Q-learning to improve its learning stability, for better comparison, we also enhanced both DQNs with double Q-learning, given that it is a simple enhancement without additional computational costs. Second, specific to ANGC's Cart Pole DQN, we found that the agent's learning has very high variance with the given hyperparameters, and the agent does not consistently achieve high performance. To counter this, we decreased the learning rate from the provided $5e^{-4}$ to $2.5e^{-4}$, and increased the target network update frequency (referred to as $C$ in [53]) from 128 to 500, which is CleanRL's value. This improved both the agent's stability and average return.

Our results in Figure 8 show that AD achieves strong performance across all 4 tasks, learning more stably and achieving higher average episodic return than the DQNs in all environments. Most notably, in comparison to DQN, AD more consistently reaches the maximum 500 return in Cart Pole, and achieves significantly higher return in Mountain Car. It also surpasses DQN's performance slightly in both Lunar Lander and Acrobot. These results suggest that AD is highly competitive against ANGC in these tasks. However, we note that as ANGC and AD learn with different signals, they are not competing approaches, and may possibly be integrated to achieve greater performance– we also look forward to exploring this potential in future work.

## D   Bootstrap Confidence Intervals

We show the results of our main experiments with 95% bootstrap intervals in Figure 9, following the recommendations by [55] to encourage better comparison of RL algorithms and reproduction of their results. For the MinAtar environments, we also show the ablation study results with 95% bootstrap intervals.

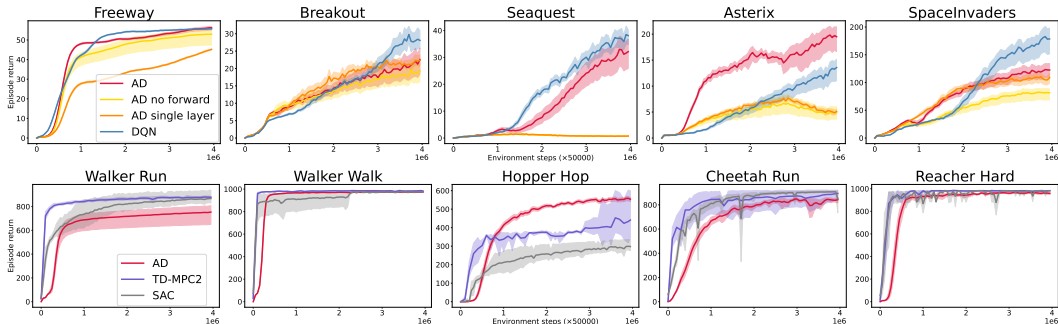

Figure 9: Episode return across the Minatar and Mujoco environments. Shaded area shows a 95% bootstrapped confidence interval around the mean computed over 20 seeds for AD and 3 seeds for TD-MPC2 and SAC respectively.

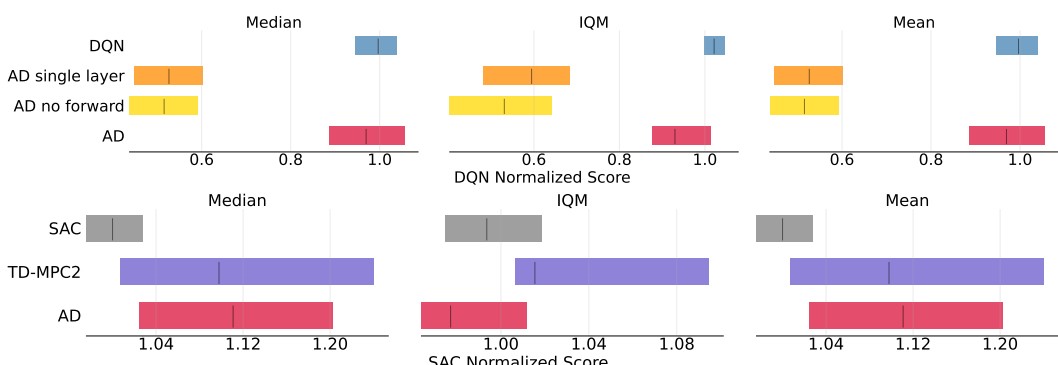

Figure 10: RLiable aggregate statistics across the Minatar and Mujoco tasks.

# E    Aggregate Statistics

Using the RLiable library provided by Agarwal et al. [1], we aggregate the performance of AD and baselines on the Minatar and Mujoco tasks. Since the MinAtar and DMC environments do not offer human-normalized scores, we used the performance of common baseline algorithms (SAC and DQN, respectively) for normalization.

We find that AD is within the confidence interval of TD-MPC2 on Mujoco and DQN on Minatar (compare Figure 10). It shows better mean and median performance than SAC on the Mujoco tasks. IQM excludes the harder hopper environment, which results in slightly worse performance of AD on this metric, as our method shines in this task.

# F    Improving Computational Efficiency in AD Cells

A limitation of our AD cells is that the $W_{\text{att}}^{[l]}$ matrix scales in the size of the action space $|\mathcal{A}|$ and the size of the hidden layer, $d$. When training on more complex environments with large action spaces and require larger hidden layers, $W_{\text{att}}^{[l]}$ can become expensive to compute and to learn. One trick we employed in DMC environments to improve the computational efficiency of our AD cells is to learn the $W_{\text{att}}^{[l]}$ matrix as a product of two smaller matrices, one $|\mathcal{A}| \times k$ and the other $k \times d$ where $k$ is a small constant. For small $k$, the number of parameters in the matrix product is significantly smaller than the original $W_{\text{att}}^{[l]}$ matrix, especially when either $|A|$ or $d$ are large, allowing for more efficient learning at the cost of expressiveness. We found that empirically in the DMC environments using $k = 8$ does not notably negatively impact performance while reducing run times.

## G  Model Hyperparameters for MinAtar

We show the hyperparameters of the AD network and DQN used in our MinAtar experiments.

| Hyperparameter | AD | DQN |
|---|---|---|
| Learning rate | $10^{-4}$ | $10^{-4}$ |
| Exploration fraction | 0.1 | 0.1 |
| Final epsilon at end of exploration | 0.01 | 0.01 |
| Loss function | Mean squared error | Mean squared error |
| Max gradient norm | 1.0 | 1.0 |
| Batch size | 512 | 512 |
| Gamma | 0.99 | 0.99 |
| Training frequency | 4 | 4 |
| Replay buffer size | $5 \times 10^6$ | $5 \times 10^6$ |
| Target network update frequency | 1000 | 1000 |

We also use a learning rate scheduler for both networks, which linearly increase learning rate to $10^{-4}$ in 500000 steps, then cosine decay until $3 \times 10^{-5}$ over remaining training steps. These hyperparameters were jointly tuned and shared by both the baseline DQN and AD network.

## H  Model Hyperparameters for the DeepMind Control Suite

We show the hyperparameters of the AD network in our DMC experiments.

| Hyperparameter | AD |
|---|---|
| Learning rate | $2.5 \times 10^{-4}$ |
| Exploration fraction | 0.25 |
| Final epsilon at end of exploration | 0.01 |
| Loss function | Mean squared error |
| Max gradient norm | 1.0 |
| Batch size | 512 |
| Gamma | 0.99 |
| Training frequency | 4 |
| Replay buffer size | $4 \times 10^6$ |
| Target network update frequency | 1000 |

## I  Environment Selection

We run our experiments on two standard RL benchmarks and 4 classic control tasks.

MinAtar [70] is an simplified implementation of 5 Atari 2600 games: Seaquest, Breakout, Asterix, Freeway, and Space Invaders. We use version 1 for all environments. The input frame size is $10 \times 10$, and the $n$ different objects in each game are placed on separate frames, resulting in a $10 \times 10 \times n$ input observation. All tasks have discrete action spaces, with up to 6 actions.

The DeepMind Control (DMC) Suite [64] is a set of low-level robotics control environments, with continuous state spaces and tasks of varying difficulty. For our experiments, we used a discretized action space, following Seyde et al. [61], as the architecture is currently only developed for discrete Q learning. From the DMC tasks we selected Walker Walk, Walker Run, Hopper Hop, Cheetah Run, and Reacher Hard. These tasks were chosen based on the size of the action space, as our discretized action heads currently scale exponentially with increasing action dimension. Using better fine-grained motor control is a valuable path for future work; for now we present the simplified variant as a proof of concept that our algorithm is able to handle a wide range of tasks and input modalities.

We chose these environments because they reflect challenges in solving complex, non-linear control tasks, from which meaningful algorithmic insight can be obtained [14], but are not so complex they require additional components like convolutional layers to solve. This makes them excellent testbeds for novel learning approaches. Although the DQN baselines published by Young and Tian [70] are

| Dataset | Hidden activations | Layer 1 | Layer 2 | Layer 3 | Layer 4 |
|---------|-------------------|---------|---------|---------|---------|
| MNIST | 500 | 0.9848 | 0.9862 | 0.9865 | 0.9867 |
| MNIST | 1000 | 0.9834 | 0.9860 | 0.9866 | 0.9870 |
| MNIST | 2000 | 0.9834 | 0.9862 | 0.9878 | 0.9874 |
| CIFAR-10 | 500 | 0.4715 | 0.5234 | 0.5394 | 0.5501 |
| CIFAR-10 | 1000 | 0.4750 | 0.5358 | 0.5496 | 0.5588 |
| CIFAR-10 | 2000 | 0.4841 | 0.5440 | 0.5610 | 0.5725 |

Table 1: Test accuracy of each layer of 4-layer AD networks trained on MNIST and CIFAR-10. Each layer has the same number of hidden activations; for example, the first row refers to an AD network with layers of 500, 500, 500, and 500 activations. Since each layer makes its own prediction, we can easily see how performance increases per layer.

obtained using a convolutional neural network (CNN), a larger DQN with only fully-connected layers can solve all tasks with comparable or stronger performance.

In addition, we provide results on the classic control tasks Cart Pole, Mountain Car, Lunar Lander, and Acrobot, which we include in Appendix C.

## J   Experiments on MNIST and CIFAR-10

To further investigate the generalizability of AD across different learning tasks, we also evaluate AD on two supervised learning datasets, MNIST and CIFAR-10, and present our results here. Q-learning is a regression problem; to adapt our architecture to solve these classification problems, we simply change the loss function to cross entropy, and make each of the `tanh` weight layers correspond to a class, instead of an action. Using a 4-layer AD network with output sizes 2000, 2000, 2000 and 2000, we achieved 98.74% test accuracy on MNIST, and 57.25% test accuracy on CIFAR-10. These results are in line with the results achieved by Forward Forward in [30].

We show our results with different sized networks in Table 1. As each layer in the network makes its own prediction, we additionally show the test accuracy of every layer. Remarkably, for almost every network size, the test accuracy steadily increases per layer. Like our experiments in RL, this supports that AD cells can learn to coordinate with each other for better performance without the need to backpropagate error signals across layers.

For clarity, we want to note that these experiments are performed simply to add another perspective to evaluate the robustness of AD's learning across tasks. AD is intentionally designed for Q-learning, rather than supervised learning, as it imitates biological processes of reward-based learning.

## K   Compute Resources

We ran our experiments on a shared scientific computing cluster using an assortment of CPU and GPUs. Each instance is run with a GPU (to improve the speed of training the neural network) and 12 CPU cores (to improve the speed of the environment simulation), and 20GB of RAM. On an Nvidia RTX 2080 GPU, a full training run of AD takes approximately 5 hours on the MinAtar environments, and 3 hours on the DMC environments. On an Nvidia A100 GPU, the run takes approximately 3.5 hours on MinAtar, and 2.5 hours on DMC. We additionally expended compute resources for hyperparameter tuning and ablation studies. In total, our experiments consumed approximately 9000 hours of compute using the aforementioned instances.

While we did not prioritize computational efficiency in the current design of our algorithm, we note that like Chen et al. [15], Lillicrap et al. [38], Journé et al. [34] and other biologically-plausible deep learning algorithms that do not suffer from the weight transport and update locking problems, our algorithm can be adapted for neuromorphic hardware that keep data local to computational units [33]. These hardware have the potential for significant energy efficiency gains.

## L    Usage of Existing Assets

We utilize several existing assets in both the implementation of our algorithm and our experiments. In Table 2 we provide their versioning, licensing and and URL for easier reproduction of our results.

| Asset | Version | License | URL |
|---|---|---|---|
| Jax [12] | 0.4.11 | Apache 2.0 | https://github.com/google/jax/ |
| MinAtar [70] | 1 | GPL-3.0 | https://github.com/kenjyoung/MinAtar |
| DMC [64] | 1.0.16 | Apache 2.0 | https://github.com/google-deepmind/dm_control/ |
| Gymnasium [65] | 0.28.1 | MIT | https://github.com/Farama-Foundation/Gymnasium/ |

Table 2: Existing assets used.

## M    Social Impacts

Our work introduces a new learning algorithm that computationally demonstrates how the credit assignment problem may be solved in the nucleus accumbens, where reward prediction errors are synchronously distributed. This research direction may yield significant positive impacts to healthcare and the social sciences. By improving our understanding of biological reward-based learning, our research has the potential to inform novel therapeutic strategies for neurological disorders characterized by dysfunctional reward processing mechanisms. By better aligning AI with biological intelligence, we also provide social science with more faithful AI agents for simulating and studying human social behavior, a practice that is increasingly adopted.

However, aligning neuroscience and AI is not without risk of negative social impacts. The development of more faithful AI agents may encourage social sciences to increasingly replace human subjects, as AI agents may be cheaper or more malleable to certain experimental setups, and exogenous variables may be more easily controlled for. Yet AI agents are inherently biased by the data they are trained upon; replacement of human subjects may further marginalize underrepresented communities, which is already a large challenge in AI fairness research.

In addition, deep reinforcement learning research, such as our work, consumes considerable amounts of computational power and energy (see Appendix K). While such research often yields valuable insights into intelligence, these gains should always be balanced against the carbon emissions caused by large-scale experiments.

## N    Discussion on Seaquest

In Seaquest, where the performance difference between 3-layer AD, AD without forward-in-time connections, and single layer AD is most apparent, the latter agents' struggle may be attributed to the failure of learning the resurfacing mechanism, which traps it at a local minima. In this environment, the agent must regularly resurface while carrying a diver to replenish oxygen. In the short term, the agent can acquire more reward if it prioritized attacking enemies, rather than managing oxygen, but if it runs out of oxygen the episode ends. We observed that the AD agent without forward connections and the single layer agent both struggled to manage oxygen, often resurfacing randomly or without first acquiring a diver. In contrast, the AD agent with the forward connections learns to manage their oxygen more optimally, and often sacrifices short-term rewards to maintain higher oxygen. In fact, during some trials the AD agent became ultraconservative with oxygen, which led to the agent surviving on average over 1500 timesteps per episode over several thousand episodes before it changed strategy. Interestingly, this same behavior was not observed in DQN agents, whose average episode lengths are consistently under 800. This suggests that AD has the potential to learn completely different policies from DQNs.

# O   Additional Related Work

Biologically inspired learning algorithms seek to reconnect machine learning with neurobiology. This supports better transdisciplinary collaboration across ML and neuroscience, which may be essential to advancements in both biological and artificial intelligence research [71].

Similar to our work, Ororbia and Mali [53] proposes a deep Q-learning algorithm, ANGC, based on the Neural Generative Coding framework [51], which is centered around the idea of active inference [54]. We indirectly compare the performance of AD and ANGC in Appendix C. Also closely relevant, Guerguiev et al. [25] proposes a BP alternative that solves credit assignment for supervised learning tasks, and show promising results on the MNIST dataset [22]. They point out that current neuroscience lacks sufficient understanding of how credit assignment is performed in the brain, and provide a method to train neural networks using segregated dendrites.

Many other related works focus on other aspects of biological plausibility in deep learning; we mention a few here for the interested reader. Lillicrap et al. [39], Lillicrap et al. [38], Kunin et al. [36], Akrout et al. [3] and Lee et al. [37] proposed algorithms that tackle the weight transport problem and non-locality of errors, which have long plagued backpropagation [24]. Journé et al. [34] adopts a Hebbian learning approach to address weight transport, locality and update locking. Miconi et al. [43] improves the plasticity of neural networks, inspired by the brain's mechanisms of neuromodulation.

