# OpenReview forum: "Temporal-Difference Learning Using Distributed Error Signals"
_NeurIPS.cc/2024/Conference — NeurIPS 2024 poster_

### Official Review · Reviewer_dpw2 · 2024-06-21

**Soundness:** 4
**Presentation:** 4
**Contribution:** 3
**Rating:** 8
**Confidence:** 4

**Summary:**

In the paper, the authors propose a novel reinforcement learning algorithm. The algorithm is based on Q-learning and uses a  biologically inspired design to work with local error signals and updates, eliminating the need for the biological implausible backpropagation. The proposed algorithm, Artificial Dopamine, is motivated by the dopamine based learning system within the brain, concretely the ventral tegmental area (VTA) and the nucleus accumbens (NAc). It is realized with a novel recurrent layer/cell type, that predicates Q values from local error signals with an attention-like mechanism. The algorithm is evaluated against several baselines in a set of established RL benchmarks, and additional ablations add further insights on the algorithm.

---
**Post-Rebuttal/Discussion**: During discussion with the authors and fellow reviewers, open questions were tackled, and additional results were presented. I've increased my score to 8 due to this.

**Strengths:**

The paper tackles an interesting and important research direction with biological plausible reward-based learning, linking computer science advancements and neuroscientific findings nicely. The approach is motivated nicely, both through the background in neuroscience as well as with the connection to the recently proposed forward-forward algorithm. Similar to that, the proposed algorithm aims at training artificial neural networks without the biological implausible backpropagation.

The method is introduced and explained understandable, with details necessary to understand the algorithm conceptually as well as to implement it.

The included figures are of high quality and are helpful illustrations.

The presented evaluations are comprehensive, solid, and appropriate for the motivating research question. Especially the provided ablations answer questions occurring while reading the paper proactively.

While not covering all aspects of biological plausible learning (as mentioned by the authors), the paper shows an elegant and competitive alternative to backpropagation that is grounded in neuroscientific insights on biological plausible reward-based learning.

**Weaknesses:**

While the majority of the paper is well written and understandable, one particular details is unclear and different sections seem to be contradicting. The AD cells/layers use multiple tanh layers, one for each action as stated and visualized. In Section 3.1, however, there is only one weight matrix W_att mentioned and used. Also in Figure 3 there is only the Q_t value, independent of an action. Inside each AD cell, are there multiple layers, one for each action, and if so, how are they used together for Q? Or is it one layer/matrix, with one dimension per action? (The first seems more plausible, but clarification and unification of notation and description would be beneficial.)

While computational demanding and, hence, understandable, 10 runs are quite low for comparing RL algorithms (known problem in the community, although often neglected/ignored). Recent frameworks and metrics have been proposed tackled at these problems related to low number of runs and comparing overall algorithms performance [1]. It would be beneficial to add such metrics (e.g., performance profiles and IQM/IQR) to the paper for a better comparison of the proposed algorithm.

------------

[1] Agarwal, Rishabh, et al. "Deep reinforcement learning at the edge of the statistical precipice." Advances in neural information processing systems 34 (2021)

**Questions:**

The algorithm is based on Q-learning, could the main ideas be transferred to other RL backbones?

The reward needs to be globally available to all layers to compute the local prediction errors. Which implications does this global target induce? Any theoretical or practical drawbacks?

How crucial is the choice of the two activation functions in the AD cell?

Typo (?) Line 101 mentions 15 tasks, while the rest of the paper mentions 14. Additionally, the main paper actually ‘only’ uses 10 tasks, the additional 4 are only provided in the Appendix for comparison with an additional baseline. Can this be clarified or better, unified in the main paper?

**Limitations:**

Limitations are discussed thoroughly in a separate Section. While detailed, they focus solely on the biological background, not mentioning technical/practical limitations.
Some of such limitations, like only working with discrete/discretized action spaces and the difficulty in scaling to larger action spaces, are mentioned in other parts of the paper, it would be good to add such practical limitations and assumptions in the dedicated limitation section.

---

> ### Author Rebuttal · Authors · 2024-08-07
>
> We sincerely thank the reviewer for their constructive feedback and detailed questions, and we’re glad to hear that you appreciate the novelty, clarity, and elegance of our work.
>
> To address your specific questions and concerns:
>
> **Clarifications on Section 3.1.**
>
> We understand how this may be confusing. Conceptually, the Q_t function takes two parameters, the state and the action, and outputs the value for that state-action pair. There exists multiple tanh layers, one for each action; W_att represents the weights of one of these layers.
> In practice, for parallelization, like [1], the Q_t function is implemented to output a vector of |A| values (where A is the action space), i.e. one value for each possible action. Each tanh layer is used to compute the value of one corresponding action. We then take the argmax over that vector to determine the action to take. We see how Figure 3 is misleading; we will update it to show that the cell computes multiple Q_t values, one for each action.
>
> **10 runs are quite low for comparing RL algorithms; additional metrics like IQM/IQR would be beneficial.**
>
> While we agree that our results can be further bolstered with more runs, RL experiments are indeed computationally demanding (Appendix H), and like most researchers, we are constrained by available resources and environmental concerns. However, we hope that the number of tasks we evaluate our algorithm on helps alleviate some of your concerns; AD is evaluated on 14 standard RL tasks, with 10 runs on each task.
>
> Further, we agree that additional metrics like IQM/IQR are beneficial for comparing the performance of RL algorithms. We’re working on additional figures that show the IQM and IQR of our data for our main results and ablation study, and will upload these before the end of the discussion period.
>
> **The algorithm is based on Q-learning, could the main ideas be transferred to other RL backbones?**
>
> Yes, we believe they can, although we haven't yet experimented with this. We chose Q-learning as its one of the most supported hypotheses for how the brain performs reward-based learning, however the same learning principles should transfer to other RL paradigms like actor-critic models.
>
> **The reward needs to be globally available to all layers to compute the local prediction errors. Which implications does this global target induce? Any theoretical or practical drawbacks?**
>
> Just making the reward signal globally available to all layers does not incur any notable drawbacks we are aware of; there is evidence that this occurs in biological learning (Section 2.1), and providing such a signal to each layer does not add any significant computational costs for modern hardware (in fact, it improves parallelization, as each layer’s update can be computed in parallel, Section 3).
>
> However, there is a significant drawback to replacing the sequentially propagated error signal with just per-layer reward and prediction errors, namely the layers can no longer directly coordinate their learning. Specifically, the upper layers of the network can't influence the learned representations of the lower layers via sequentially propagated gradients. The problem with gradient propagation is that there is no evidence it in the brain, despite its known importance in deep learning. This is partially why we believe our results are surprising and significant: we show that just distributed, local prediction errors and updates may already be sufficient to learn many complex reward-based learning tasks, without the need to directly coordinate updates across layers during learning.
>
> **How crucial is the choice of the two activation functions in the AD cell?**
>
> We experimented with alternative choices for these two activation functions, including replacing the relu function with a leaky relu, and replacing the tanh function with a sigmoid and with a second relu function. Most combinations of different activations did not have a significant impact on performance, with the exception of using two relu functions, which made learning unstable.
>
> For transparency, our experiments on activation function choice were not extensive; there could certainly be combinations of other activation functions that lead to better performance.
> However, as our primary goal is not to achieve state-of-the-art performance, we believe that any potentially unrealized gains in performance does not interfere with our contributions, and thus leave further hyperparameter tuning of AD cells to future work.
>
> **Line 101 mentions 15 tasks, while the rest of the paper mentions 14; the main paper uses 10 tasks.**
>
> We agree this can be further clarified. Line 101 is indeed a typo; we evaluated AD on 14 tasks, including 5 tasks from the MinAtar testbed, 5 tasks from the DeepMind Control Suite, and 4 classic control tasks. We relegated the 4 classic control tasks to Appendix C because they are relatively less challenging compared to the MinAtar and DMC tasks (despite still being an informative starting point for evaluating the performance of reward-based learning algorithms). As we achieved consistently strong performance on each of these 4 tasks, to avoid cluttering the main paper body (given the space limit), we moved them to the appendix.
>
> To address this issue, we have fixed the typo, and updated the Experiments section (Section 4) to explicitly state the number of tasks.
>
> **While technical and practical limitations are mentioned in other sections, the Limitations section only discusses limitations related to biology.**
>
> We will add a dedicated paragraph in the Limitations section to discuss technical and practical limitations, including discretized action spaces, current difficulties with scaling, and the number of runs per task.
>
> ---
> **References**
>
> [1] Mnih, Volodymyr, et al. "Human-level control through deep reinforcement learning." *nature* 518.7540 (2015): 529-533.

---

> > ### Comment · Reviewer_dpw2 · 2024-08-09
> > **Thanks.**
> >
> > Thank you for your detailed answer to my review and questions. The answers clarified the open questions and I hope to see the mentioned text changes in a potential final paper.
> > Also waiting for "We’re working on additional figures that show the IQM and IQR of our data for our main results and ablation study, and will upload these before the end of the discussion period." for a better comparison of the methods.

---

> > > ### Comment · Area_Chair_aQyN · 2024-08-13
> > >
> > > Hi reviewer dpw2 and authors,
> > >
> > > dpw2: Just to clarify, when you say "Also waiting for...," do you think these additional figures are critical for the paper's completeness? That is, would your overall assessment go down if the authors did not offer these additional figures?
> > >
> > > Authors: Are you able to provide the IQM and IQR from your experiments?

---

> > > > ### Author Response · Authors · 2024-08-13
> > > >
> > > > Thank you, Reviewer dpw2, for your positive response and for taking the time to review our answers. We also appreciate Friendly Neighborhood AC for checking in on us!
> > > >
> > > > We’re pleased to share that we have now generated the IQM/IQR plots (apologies for the delay—our resident RL expert is traveling): https://anonymous.4open.science/r/AD_neurips_rebuttal-3723/IQM.pdf
> > > >
> > > > Regarding the DMC environments, the data for the SAC and TD-MPC2 baselines (taken from standard implementations by Denis Yarats & Ilya Kostrikov, 2020, and Hansen et al., 2023, respectively) had fewer seeds, which made individual environment comparisons less meaningful due to the low number of runs. Therefore, we opted to aggregate performance across the MinAtar tasks and the DMC tasks and present the aggregate statistics. Since the MinAtar and DMC environments do not offer human-normalized scores, we used the performance of common baseline algorithms (SAC and DQN, respectively) for normalization. Our plots display the IQM (with range), median, mean, and optimality gap.
> > > >
> > > > Additionally, to demonstrate that our performance is consistent across each individual tasks, we computed 95% bootstrap confidence intervals for each task, following the method recommended by Patterson et al. (2023) for RL best practices. These are plotted here: https://anonymous.4open.science/r/AD_neurips_rebuttal-3723/boostrapped%20CI.pdf
> > > >
> > > > We will ensure that the style and formatting of these plots align with the other figures in our paper, and we will include them in the camera-ready version if accepted. Please let us know if you have any further questions or concerns.
> > > >
> > > > ---
> > > >
> > > > **References**
> > > >
> > > > Denis, Y. and Kostrikov, I. (2020). Soft actor-critic (sac) implementation in pytorch. https: //github.com/denisyarats/pytorch_sac.
> > > >
> > > > Hansen, N., Su, H., & Wang, X. (2023). Td-mpc2: Scalable, robust world models for continuous control. arXiv preprint arXiv:2310.16828.
> > > >
> > > > Patterson, A., Neumann, S., White, M., & White, A. (2023). Empirical design in reinforcement learning. arXiv preprint arXiv:2304.01315.

---

> > > > > ### Comment · Reviewer_dpw2 · 2024-08-13
> > > > >
> > > > > Thank you for the additional effort in clarifying concerns and providing additional results. There are no open issues from my side and I still think the paper is a valuable contribution to the conference.
> > > > >
> > > > > @AC: Also thanks to you for your engagement! I'd wish it would be like this on all papers...

---

### Official Review · Reviewer_8Waq · 2024-07-03

**Soundness:** 4
**Presentation:** 3
**Contribution:** 3
**Rating:** 7
**Confidence:** 4

**Summary:**

This paper addresses the distributed credit assignment problem in biological reinforcement learning, where a naive implementation of backpropagation is implausible. The authors show that it is possible to update action values using a variant of the forwward-forward algorithm specialized for RL. They then validate that this algorithm is effective on a range of standard benchmarks.

**Strengths:**

- The problem (how to implement scalable deep RL in a biologically plausible way) is important.

- The paper is clearly written, with a good review of both the biology and AI background.

- The results are impressive given that the system does not use backprop.

- I appreciated the careful ablation analysis.

**Weaknesses:**

- p. 5: The review by Glimcher (2011) is cited to support the claim that NAc (ventral striatum) neurons signal action value. In fact, there is a long tradition arguing that it is actually the dorsal striatal subregions (caudate and putamen) which signal action values, whereas ventral striatum signals state value (i.e., a form of actor-critic architecture; see Joel et al., 2002; O'Doherty et al., 2004). Most studies I'm aware of have shown that signals related to action value are relatively weak or absent in ventral striatum (see for example Ito & Doya, 2015).

- Overall I felt that the biological evidence for the modeling framework was weak. What is the evidence that dopamine updates values locally in the NAc? This is physiologically tricky because dopamine acts postsynaptically through volume transmission, diffusing broadly in the extracellular space. What is the evidence for the particular form of hierarchical message passing architecture proposed here? Basically, I felt that the ideas here are interesting in principle but as presented are rather disconnected from empirical data.

Minor:

- p. 2: "connection ," -> "connection,"

**Questions:**

- I was unsure whether the claim here is that the NAc is a multi-layer network or one layer in a multi-layer network.

- If the authors would like their model to be taken seriously by neurobiologists, they need to do more to directly link it to empirical evidence.

- Ideally the authors could make some neural predictions based on the model. It would also be helpful to understand how the predictions are different from those of other models.

Rebuttal update: I have raised my score from 6 to 7 following the authors' responses to my comments.

**Limitations:**

The authors extensively discuss limitations in section 6. There are no potential negative societal impacts.

---

> ### Author Rebuttal · Authors · 2024-08-07
>
> We sincerely thank the reviewer for their insightful feedback, and we’re grateful for the opportunity to improve and clarify our work through answering your detailed questions and concerns.
>
> **There is a long tradition arguing that it is actually the dorsal striatal subregions (caudate and putamen) which signal action values, whereas ventral striatum signals state value.**
>
> We acknowledge that there are multiple theories that provide different mappings of reinforcement learning frameworks to the brain’s reward circuitry. We want to clarify that our assumptions and findings are not in conflict with the actor-critic models by Joel et al. (2002) and O’Doherty et al. (2004). Actor-critic mappings of the reward circuitry, like Joel et al. (2002) and O’Doherty et al. (2004), argue the dorsal striatum is responsible for action selection, i.e. policy learning, whereas the ventral striatum is responsible for value learning. On the other hand, Q-learning is a simpler model that combines policy and value learning. The main difference (with respect to our work) is whether action selection occurs separate or together with value learning.
>
> AD does not make the argument that action selection need happen in the NAc. Rather, we just take the assumption that the NAc computes the Q-value function, which takes in the state and action as parameters, and performs Q-learning. The action can be provided from another region like the dorsal striatum. This view is consitent with the findings of works by Roesch et al. (2009), Mannella et al. (2013), and Aston et al. (2024).
>
> On the other hand, we acknowledge that there exists research like Ito and Doya (2015) and Weglage et al. (2021) that suggest the competing theory that the dorsal striatum is responsible for signaling action values, whereas the ventral striatum signals state values. For completeness and openness, we will include this in the Limitations section.
>
> **The biological evidence for the modeling framework was weak. What is the evidence that dopamine updates values locally in the NAc, and for the particular form of hierarchical message passing architecture proposed here?**
>
> The theory that dopamine updates values locally in the NAc is supported by Wightman et al. (2007), who found spatial and temporal heterogeneity of dopamine release within the NAc. This suggests that while dopamine concentration is homogeneous within each microenvironment containing a local subpopulation of NAc neurons, it can vary across different subpopulations.
>
> The hierarchical message passing architecture is a design choice inherited from current deep learning practices, reflecting the passing of information between subpopulations of neurons. Our specific architecture is not intended to be a fully mechanistically accurate model of the NAc, but rather a computational model to demonstrate that it is possible to learn complex reward-based tasks using synchronously distributed, locally homogeneous error signals. The actual implementation in the NAc may differ. This counterintuitive and surprising finding potentially opens new avenues for explaining biological reward-based learning,  which is why we argue that we make a valuable contribution.
>
> **I was unsure whether the claim here is that the NAc is a multi-layer network or one layer in a multi-layer network.**
>
> Each layer in our model represents a local subpopulation of NAc neurons that receive the same signals from a subpopulation of dopamine neurons. However, as our model is not intended as a mechanistically accurate representation of the NAc, we do not claim that the NAc is a multi-layer network.
>
> **If the authors would like their model to be taken seriously by neurobiologists, they need to do more to directly link it to empirical evidence.**
>
> We agree that additional biological connections can strengthen our work. However, the main contribution of our work is not to propose a biological model, but to show that synchronously distributed, locally homogeneous error signals observed in the mesolimbic dopaminergic system may be sufficient to teach neurons complex reward-based tasks. While we make some biological assumptions for modeling (e.g., NAc neurons encode approximations of the value function, and VTA projections may compute errors specific to these approximations), we do not claim our work proves these assumptions; they are part of our premise, not our hypothesis.
>
> We demonstrate that if these assumptions hold true, such conditions may be sufficient for complex reward-based learning by neural networks, without the need for other mechanisms to explicitly coordinate credit assignment. Previously, it was believed that solving complex nonlinear tasks required neurons in different regions to sequentially pass error signals and explicitly coordinate their learning. This assumption is held by both biologically plausible local-learning algorithms and backpropagation. We show that this assumption may not be necessary.
>
> **Ideally the authors could make some neural predictions based on the model. It would also be helpful to understand how the predictions are different from those of other models.**
>
> We agree that making neural predictions based on our model is an important direction. As an initial step towards better alignment with neurobiology, we performed experiments showing agreement with the distributional dopamine coding model by Dabney et al. (2025) in Section 5. We plan to further develop this aspect of our work in the future.

---

> > ### Comment · Reviewer_8Waq · 2024-08-07
> > **thanks**
> >
> > Thanks for your thorough response to my comments. I hope that at least some of this will be reflected in the paper.

---

> > > ### Author Response · Authors · 2024-08-12
> > >
> > > Thanks for acknowledging our rebuttal. We appreciate your feedback and are committed to reflecting your suggestions in the revised paper.
> > >
> > > To address your concerns, we plan to make the following updates:
> > >
> > > 1. Clarify that AD does not require action selection to occur in the NAc and acknowledge competing theories (Ito & Doya, 2015; Weglage et al., 2021) in the Limitations section.
> > > 2. In the background section, clarify that dopamine providing local signals for updates is supported by the findings of Wightman et al. (2007).
> > > 3. Add to the Limitations section that our hierarchical message passing architecture is a design choice from deep learning practices and not a fully mechanistic model of the NAc.
> > >
> > > We'd value your feedback on whether these updates address your concerns and if there are any other critical points we should prioritize to potentially raise the score.
> > >
> > > Thank you again for your constructive feedback.

---

> > > > ### Comment · Reviewer_8Waq · 2024-08-12
> > > > **raising score**
> > > >
> > > > Yes, I will raise my score from 6 to 7.

---

> ### Author Response · Authors · 2024-08-07
>
> **References**
>
> Roesch, M. R., Singh, T., Brown, P. L., Mullins, S. E., & Schoenbaum, G. (2009). Ventral striatal neurons encode the value of the chosen action in rats deciding between differently delayed or sized rewards. *Journal of Neuroscience*, *29*(42), 13365-13376.
>
> Mannella, F., Gurney, K., & Baldassarre, G. (2013). The nucleus accumbens as a nexus between values and goals in goal-directed behavior: a review and a new hypothesis. *Frontiers in behavioral neuroscience*, *7*, 135.
>
> Ashton, S. E., Sharalla, P., Kang, N., Brockett, A. T., McCarthy, M. M., & Roesch, M. R. (2024). Distinct Action Signals by Subregions in the Nucleus Accumbens during STOP–Change Performance. *Journal of Neuroscience*, *44*(29).
>
> Weglage, M., Wärnberg, E., Lazaridis, I., Calvigioni, D., Tzortzi, O., & Meletis, K. (2021). Complete representation of action space and value in all dorsal striatal pathways. *Cell reports*, *36*(4).
>
> Wightman, R. M., Heien, M. L., Wassum, K. M., Sombers, L. A., Aragona, B. J., Khan, A. S., ... & Carelli, R. M. (2007). Dopamine release is heterogeneous within microenvironments of the rat nucleus accumbens. *European Journal of Neuroscience*, *26*(7), 2046-2054.

---

### Official Review · Reviewer_Sdew · 2024-07-10

**Soundness:** 2
**Presentation:** 4
**Contribution:** 2
**Rating:** 4
**Confidence:** 5

**Summary:**

In this work, the Authors aimed to propose a computational model that is consistent with classical functions assigned to select regions in the reward system in the brain. To this end, they build upon George Hinton’s Forward-Forward algorithm, extending it with the ability to robustly generate continuous (instead of categorical) values and endowing it with the DQN loss. The Authors show that the proposed algorithm reaches competitive scores on a panel of standard RL benchmarks, and discuss its potential mapping on the brain circuitry.

**Strengths:**

The text is clear and almost always straightforward. Many key statements are accompanied by reasonable disclaimers; as a result, the claims are seldom overstated. The paper has a reasonable description of the related (used) algorithms. The figures are high-quality, which makes the paper stand out among many other submissions.

The experiments are performed on a sizeable panel of benchmarks. Wherever the baseline design had to be altered to ensure a fair comparison with the proposed model, correct adjustments are performed including the (hyper)parameter search for the altered models.

There may be a subcommunity interested in these results. After George Hinton’s successful presentation of his Forward-Forward algorithm at NeurIPS 2022, many people present at this keynote lecture got interested in the topic.

**Weaknesses:**

The paper’s neuro premise (Sections 1; 2.1) reads as if the roles (or even activities) of the brain regions involved in the brain circuit are known undebatable. Even though this impression is then refuted in Section 6 (which is commendable), most of the paper operates under this assumption. While the paper correctly cites some classical literature on the topic, the debates over the brain region roles proposed in these works led to more nuanced models, e.g. the mappings of the actor-critic model on the reward circuit, and then to an overall notion that such mappings represent a bird-eye view of the problem and may not necessarily be mechanistically accurate. It would be nice to explore and discuss this uncertainty, as it may alter the ground truth for this work.

The paper’s comp neuro premise limits itself to the Forward-Forward algorithm. Although some of the other related work is mentioned in Section 7, these and other works have not made it into the model’s design. Meanwhile, the field of biologically plausible learning, including RL, is immense. There’s a host of work on predictive coding and local learning rules; the new works are submitted to (almost) every major ML conference. There are noteworthy works on the computational role of dopamine and RL in the brain, including the works on "Backpropamine" and DeepMind’s modeling of distributional dopamine coding. In that sense, it’s unclear why the current submission focuses on a single model, mostly validated within the ML community while ignoring the host of models validated within the computational neuroscience community. This may be mitigated by the Authors’ comment that their goal was to present a model consistent with neuroanatomy, which is valid, but brings me to the next point.

A single model consistent with data doesn’t preclude the existence of multiple other models, potentially with different mechanics, leaving the chance that the proposed model has little to do with the biological reality. Indeed, a Monte Carlo – or even brute force model – will eventually learn an approximately correct Q-function and won’t need backprop / coordinated feedback signal to do so. While – and I would like to stress that – the results in this paper are correct and the wording is correct as well, arguably, to learn something from this result, one would need to perform multiple additional comparisons between the (versatile, detailed) neuroanatomical structure of the reward circuit and a panel of prominent computational models. As such comparison is not performed within the scope of this work, it is arguably too early to recommend this paper for publication in NeurIPS at this point.

**Questions:**

N/A

**Limitations:**

Related literature on neuroanatomy and computational models of the reward system are not considered.

The proposed model is not contrasted against other candidate models and thus may be uninformative.

---

> ### Author Rebuttal · Authors · 2024-08-06
>
> Thank you for the review. We noticed that this review is nearly identical to an earlier review we received at a different conference. Building on the previous feedback, we have significantly updated our manuscript with additional experiments, ablation studies, and expanded discussions, including new experiments and algorithmic updates aligning with DeepMind’s distributional dopamine coding (Section 5). We are thus surprised to receive the same review with a decreased score. Assuming you are the same reviewer, we wonder if there might have been a transpositional error or if the old version of our manuscript was reviewed by mistake?
>
> In either case, we’d like to directly respond to your concerns, and highlight the relevant updates we made since our last submission that aims to address these concerns.
>
> **The neuro premise in Sections 1 and 2.1 reads as if the roles of the brain regions involved in the brain circuit are known undebatable. This is refuted in Section 6, but the discussion is inadequate.**
>
> We acknowledge this was a weakness in our previous submission. We expanded our Limitations section to explicitly state the assumptions we make regarding the biological mapping and included additional discussions on alternative mappings of RL frameworks to the reward circuit. We additionally discussed works by Takahashi et al. (2008), Chen & Goldberg (2020), Morris et al. (2006), Niv (2009), Roesch et al. (2007), Akam & Walton (2021), and Coddington et al. (2023). We highlighted that these theories are high-level mappings and may not be mechanistically accurate. We also added a footnote and forward reference to ensure this discussion is not overlooked.
>
> **The paper's comp neuro premise limits itself to the Forward-Forward algorithm, and ignores the host of other models in the field of biologically plausible learning.**
>
> Our main investigations revolve around an architecture inspired by the Forward-Forward algorithm. This does not mean we disregard other models in biologically plausible learning. Instead, our focus was to address a specific problem in biological reward-based learning: how neurons can coordinate learning to solve complex, nonlinear RL tasks using regionally homogenous, synchronously distributed error signals. While there are many related methods, they address different aspects of biologically plausible learning.
>
> Taking a reductionist approach allows us to tackle more complex modern RL tasks, highlighting what TD-learning with distributed error constraints can achieve relative to non-constrained current deep RL algorithms. This approach helps attribute performance differences to distributed errors and the downsides of our algorithm, without being confounded by other shortcomings in current biologically plausible deep learning.
>
> We agree that integrating our model with other established methods, like DeepMind’s model of distributional dopamine coding (Dabney et al., 2020), is important. Since our last submission, we designed an extension of our algorithm based on the quantile regression method to learn distributions over values (Section 5). Our goal is to evaluate whether AD can learn such distributions, aligning with Dabney et al.’s work suggesting the brain’s value predictors learn distributions over values, rather than the mean value. Like Dabney et al., we use quantiles to reflect differences in optimism/pessimism across dopamine neurons. This distributional version of AD consistently achieved strong performance on DMC tasks, though it was slightly less sample efficient than the standard version of AD.
>
> We have not yet integrated with Backpropamine (Miconi et al., 2020), which focuses on neuromodulated plasticity and is less directly relevant to our problem, but we we look forward to potentially exploring this direction in future work.
>
> Finally, we expanded our related work section (Appendix N) to include additional works in biologically plausible learning.
>
> **A single model consistent with data doesn’t preclude the existence of multiple other models, potentially with different mechanics, leaving the chance that the proposed model has little to do with the biological reality.**
>
> We agree that our model solving the credit assignment problem in reward-based learning doesn't preclude other models from doing so. Our goal is not to claim our model is the only explanation, but to provide one potential solution to this problem, which, to our knowledge, has been lacking.
>
> To our knowledge, no published works consistently solve all MinAtar tasks or our subset of DMC tasks without backpropagation; we are the first to do so. Ororbia & Mali (2022) proposed a similar backprop-free Q-learning algorithm, but their evaluations are limited to simpler tasks. We present results on 3 of these tasks in Appendix C (the fourth was not publicly available).
>
> We also want to address the biological reality issue, and explain why we believe our work is biologically relevant.
> The central hypothesis of our work is that synchronously distributed, locally homogeneous error signals, observed in the mesolimbic dopaminergic system, may already be sufficient to teach neurons to solve complex reward-based learning tasks. While we assume that NAc neurons encode approximations of the value function (and that VTA projections may compute errors specific to these approximations) in our experiments, we do not claim that our work proves this to be true; this assumption is part of our premise, not our hypothesis (we agree that alternative theories exist).
>
> Rather, our findings suggest that if our assumptions hold, such conditions may be sufficient for complex reward-based learning by neural networks without the need for mechanisms explicitly coordinating credit assignment. This is counterintuitive and challenges established beliefs, potentially opening new avenues for explaining biological reward-based learning, and thus we believe it is a meaningful contribution.

---

> > ### Comment · Reviewer_Sdew · 2024-08-08
> >
> > I would like to thank the Authors for their detailed response. Below, I address the Authors' specific concerns and elaborate on my report.
> >
> > Regarding the similarity to the previous review, I would like to describe the way I've prepared this one to clear all possible concerns in this aspect. First, I've compared the new version of the paper with the old one. To do that, I went through the main text of each paper, comparing them paragraph by paragraph. What I found is that, besides the minor formatting changes aimed at accommodating the NeurIPS format, there were three paragraphs added: the first paragraph in Section 4, the last paragraph in Section 5, and the middle paragraph in Section 7. Second, I went through my old review to see the points that I still view as valid, considering the added paragraphs and the extended discussion with the Authors last time, which has provided me with a lot of important details regarding this work (the Authors' engagement was much appreciated). The added experiments did not contrast the proposed model with other existing models, and the disclaimers on the biological plausibility, while more abundant now, were still sparse. Finally, to determine my score, I've factored in my previous assessment of this work and the scope of the changes made between the two versions. Specifically, while the previous rebuttal period was short, limiting the scope of changes that could be introduced during it, there's much more time between the submissions, so typically substantial revisions of the papers are expected.
> >
> > Now to the score part. This was (and is) one of the harder submissions I've got to review (and score) for the following reasons. On one hand, the paper is technically correct. All appropriate disclaimers are made regarding the biological relevance of the work. On the other hand, the biological relevance is constantly alluded to throughout the text, potentially leading a reader to the idea that the paper explains how the brain works, and likely leading the reader to the conclusion that the paper has a high impact on the field of biology. Notably, most of the disclaimers and additional considerations are provided at the end of the main text and in the Appendix, way after the readers' opinions on the work are formed. This concern is consistent with the reports of all other Reviewers here who have expressed concerns regarding the biological relevance of this work and/or the language describing such.
> >
> > What I would see as a proper way to present this work in its current scope is to be more upfront about the limitations, such that the reader would immediately know that the mechanistic modeling of the brain was not a goal or a result of this work, but rather a new computational architecture was tested, inspired by some neuroscience works. Then, the paper could be assessed based on its direct contribution rather than on what some may see as an alluded promise. The algorithm itself is a valid contribution and, as such, should absolutely be published somewhere.
> >
> > Not that it matters, but I would be happy to revise my score to borderline contingent on the changes in the message from biology-focused to algorithm-focused. I'm happy to re-read the updated anonymous PDF with such changes. Please also let me know if there are other changes or considerations that I may have overlooked.

---

> > > ### Author Response · Authors · 2024-08-12
> > >
> > > Thank you for engaging. We have made several substantive improvements in the paper since our last submission that we’ll highlight below, and we’ll edit the paper in revision to make these changes more clear. Some additional figures were in the new appendices, but if important can be moved into the main text.
> > >
> > > 1. **Algorithm Extension**: We designed and evaluated an extension of our algorithm to learn distributions of values, aligning our work with Dabney et al. (2020) (Section 5 & Appendix M).
> > > 2. **Sample Efficiency**: We significantly improved the sample efficiency of our algorithm in the DMC environments (Section 5).
> > > 3. **Ablation Studies**: We conducted additional ablation studies on DMC and MinAtar environments, analyzing the effects of varying layer sizes in a single-layer AD network (Section 5 & Appendix L).
> > > 4. **Biological Assumptions**: We expanded our discussion on biological assumptions and alternative hypotheses, citing eight additional neuroscience works (Section 6), and added a forward reference to address your concern about potentially misleading readers.
> > > 5. **Hypothesis Falsifiability**: We defined the criteria of sufficiency to explicitly make our hypothesis falsifiable (Section 4).
> > >
> > > We agree that our work is more algorithm-focused, and we do not claim that our model is a realistic biological reconstruction of specific brain mechanisms. However, our motivation for developing this algorithm is rooted in biological observations, and we believe our contribution holds relevance for the field. Specifically, our paper focuses on an algorithmic insight relevant to biology—demonstrating that distributed error signals, as observed in the brain, can be sufficient for reward-based learning—rather than on biological modeling per se.

---

> > > > ### Comment · Area_Chair_aQyN · 2024-08-13
> > > >
> > > > Hi reviewer Sdew,
> > > >
> > > > We are about to lose access to the authors (August 13, 11:59pm AoE). Let's try to clear up as many points of contention as we can before then.
> > > >
> > > > From your comment it sounds like your primary concern is that the paper may still give the impression that it is presenting a model of biological processes rather than, per the authors stated intent, a biologically inspired algorithm that demonstrates the feasibility of a hypothesized biological phenomenon. Is this accurate or are there other critical issues that have not been addressed by the additions to the paper (summarized succinctly in the authors' most recent comment)?
> > > >
> > > > Vis a vis that concern, I see that reviewer dpw2 has commented that they didn't perceive this to be a significant issue in the paper and that they felt clear about the intent and the contribution. So there is at least some evidence that some readers are not misled. Are there critical places in the paper that you feel are particularly misleading or are in particular need of clarification of this point? Perhaps in the brief time we have left, the authors could offer some alternative wording that might sharpen the intent of the paper. Do you believe that addressing this issue this would be a matter of a few well-placed, clearly stated sentences or significant revision of the entire work?
> > > >
> > > > Finally two quick points:
> > > >
> > > > 1) This paper is not in a resubmission track and extent of change from a submission to a different conference is not part of the NeurIPS acceptance criteria. It may certainly be that weaknesses of that submission have carried over, but let's please set aside "scope of the changes" in the assessment and focus on the merits of the manuscript that was submitted to NeurIPS.
> > > > 2) Considering the authors' summary of the additions to the paper, please take a look back at your review and consider whether any of the comments based on the old manuscript are no longer applicable; if so, please consider revising the review to reflect the current manuscript.
> > > >
> > > > Thanks!

---

> > > > > ### Comment · Reviewer_Sdew · 2024-08-13
> > > > >
> > > > > Dear Area Chair aQyN,
> > > > >
> > > > > thanks for doing everything to clear the remaining questions toward the informed decision on the paper.
> > > > >
> > > > > Your summary is correct.
> > > > >
> > > > > I believe that the point I'm raising is not a matter of editing a few critical places in the text but rather of a general message. I welcome the changes introduced by the Authors regarding further disclaimers on the extent of the biological plausibility of the model, however, starting from the abstract, the text has a clear biological focus - rather than an algorithmic focus - so my concerns remain. Fortunately, we have multiple Reviewers here, so my take on the text only matters that much. I also acknowledge the Reviewer dpw2's comment on not perceiving it as a significant issue (along with other Reviewers' positive reports, indicating the same assessment of the text). In that light, I'll consider revising my score, though it will take longer.
> > > > >
> > > > > To the specific points:
> > > > > 1. Resubmission. I've only mentioned it regarding the Authors' question on the previous score vs. the current score. The only point I'm making in that regard is that the previous conference score first reflected the initial uncertainty and then the time limitations of the rebuttal period while none of these are the factors here at NeurIPS, hence the different number.
> > > > > 2. While all the outlined changes are indeed welcome, they mostly address the algorithmic part of the work, thus being tangential to my concerns. Hypothesis falsifiability sounds interesting though and I'll be happy to read the related parts of the text shortly, open to revisiting my score.

---

> > > > > > ### Comment · Reviewer_Sdew · 2024-08-13
> > > > > >
> > > > > > After careful consideration, based on the Reviewers' reports and Authors' feedback, I raise my score to borderline to best reflect the current state of the work, under the NeurIPS guidelines.

---

### Official Review · Reviewer_K1xv · 2024-07-12

**Soundness:** 3
**Presentation:** 4
**Contribution:** 2
**Rating:** 7
**Confidence:** 4

**Summary:**

An novel, biologically plausible RL algorithm based on DQN and Fast-Forward called Artificial Dopamine is introduce. The algorithm is inspired by Dopamine pathways in the human brain and uses synchronous, locally computed per-layer TD errors for training. Additionally, a novel recurrent network architecture is proposed where layer outputs are used as inputs to the respective preceding layer at the next time step. The approach is evaluated on a range of discrete and continuous RL tasks.

**Strengths:**

Investigating biologically plausible learning is an interesting field of research and surely significant. The manuscript is well written and reads very smoothly. Explanations of biological phenomena are clear and informative, and foremost rigorously cited. The authors clearly distinguish which parts are inspired by biology and which are deliberate choices due to empirical reasons.  Limitations and societal impact are discussed thoroughly.

**Weaknesses:**

Minor Issues:
- L 116: "and is" -> "and are"
- L134: "in many practical applications, such as TD learning in the brain". Makes it sound like this a proven fact that there is TD learning taking place in the brain...

**Questions:**

- L252: "Since the local updates are distributed and per-layer, they can be computed in parallel." The temporal dependency between time-steps due to the recurrence is still present. So, does this actually translate to improved runtimes?

**Limitations:**

The Limitations such as the use of the use of a layered structure, and limited proof for the method implemented (Q-learning) occurring in the brain are discussed in detail.

---

> ### Author Rebuttal · Authors · 2024-08-06
>
> We sincerely thank the reviewer for their constructive feedback, and we’re glad to hear that you appreciate the significance, scientific rigor, and clarity of our work.
>
> We have fixed the typo on L116, and reworded L134 as “…in many practical applications with large or continuous state spaces…” to avoid confusion; thank you for pointing out these issues.
>
> The reviewer also raised an insightful question: whether the temporal dependency between time-steps due to recurrence would prevent parallelization (and thus improved runtimes). For most recurrent architectures trained with backpropagation-through-time, this is indeed the case; the updates of each layer at each time step must be sequentially computed. However, in AD, unlike backpropagation-through-time, we do not propagate any error signals back along the recurrent connections, thus there does not exist any such dependencies during learning. To elaborate, the forward-in-time recurrent connections only serve to pass activations from upper layers to lower layers; no error information is passed back through these connections. Therefore, the recurrent connections do not prevent parallelization.

---

> > ### Comment · Reviewer_K1xv · 2024-08-08
> >
> > Thank you for the given answer. However, it is still unclear to me how the model could be parallelized.
> >
> > Assuming you are training weights that process the last timestep's input, you will need to know the input before updating those. I guess you could buffer the intermediate layer's activations during rollout for later training, entailing a bias in the gradients when you update the model more than once on the same batch of collected data?

---

> > > ### Author Response · Authors · 2024-08-12
> > >
> > > Thanks for the follow-up question. You’re right— we can save the last timestep’s activations along with the sequences into the replay buffer, and use these values for training, but there would be a bias in the gradients as the model parameters are updated. Instead, to avoid this, we store short sequences during rollout into the memory buffer, then replay the sequences during training, zero-initializing the first "recurrent" state. This follows established practice in deep RL for training RNNs (Hausknecht & Stone, 2015; Kapturowski et al., 2018). Does that answer your question?
> > >
> > > ---
> > >
> > > **References**
> > >
> > > Hausknecht, M., & Stone, P. (2015, September). Deep recurrent q-learning for partially observable mdps. In *2015 aaai fall symposium series*.
> > >
> > > Kapturowski, S., Ostrovski, G., Quan, J., Munos, R., & Dabney, W. (2018, September). Recurrent experience replay in distributed reinforcement learning. In *International conference on learning representations*.

---

### Comment · Area_Chair_aQyN · 2024-08-08
**Discussion Period**

Hi all! Just your friendly area chair checking in.

First, thanks to the reviewers for your work so far on this paper. The discussion period has begun, which includes both reviewers and authors. This is our opportunity to clear up misunderstandings, get answers to questions, and generally gather enough information to make an informed decision grounded in the acceptance criteria.

It seems there is general agreement that problem addressed by the paper (developing biologically plausible RL algorithms at scale) is important and that the paper is well-presented. The main critiques raised seem to concern insufficient acknowledgement of alternative neuroscientific hypotheses and biological models, and the extent of the biological evidence that this model could correspond to real activitiy/mechanisms in brains. Does that seem like a fair high-level summary?

Reviewers: please carefully read the other reviews and the authors' responses (with included PDF) and let us know what follow-up questions you have and to what extent your evaluation might change as a result of the additional context. Please especially raise any points of disagreement with other reviewers or the authors, as these are opportunities for us to clarify misunderstandings and detect misconceptions.

---

> ### Comment · Reviewer_Sdew · 2024-08-08
>
> Thank you for starting the discussion. I see your summary as correct.

---

> ### Comment · Reviewer_dpw2 · 2024-08-09
>
> Thanks for this thread.
>
> In general I agree to the high-level summary, although the critique on *"the extent of the biological evidence that this model could correspond to real activity/mechanisms in brains"* is less an issue in my opinion. That was not the intend of the paper nor did the authors claim this (please correct me if I'm wrong), but rather aimed to show a *bit more biological plausible* RL algorithm that is *inspired* by neuroscience and recent ML results (ff), and that eliminates the need of BP.

---

> > ### Author Response · Authors · 2024-08-14
> >
> > Reviewer dpw2 is correct: we don’t claim that AD is a mechanistically accurate model of real brain activities. Our paper focuses on an algorithmic insight relevant to biology—demonstrating that distributed error signals, as observed in the brain, can be sufficient for reward-based learning—rather than on biological modeling per se. We propose a more biologically plausible RL algorithm that eliminates the need of BP in order to do so.
> >
> > We want to again express our gratitude towards the AC for diligently facilitating the discussions. Quoting Reviewer dpw2, we wish it would be like this on all papers.

---

### Decision · Program_Chairs · 2024-09-25

**Decision:**

Accept (poster)

**Comment:**

There is broad agreement amongst the reviewers that the biologically-inspired algorithmic contribution of the paper is interesting, novel. and significant, both because it demonstrates an RL method without backpropagation and because it builds upon constraints from hypotheses in neuroscience. The main points of contention concern the extent to which the paper presents unsettled neuroscientific hypotheses as fact and the extent to which the paper offers or claims to offer insight regarding biological systems. Especially after the discussion period, most of the reviewers agree that the paper is sufficiently clear about its intent and its claims though there is some disagreement on this point.

The overall assessment seems to be that the paper is interesting on a technical/conceptual level but that the paper could be stronger in its communication with the neuroscience community. I recommend acceptance, but I also urge the authors to carefully consider the critiques from these reviewers and, while preparing the camera-ready version, to focus on clear, up-front statements about the disputed nature of the underlying neuroscience hypotheses and about the limited nature of any biological insights.